# DAG: A Dual Correlation Network for Time Series Forecasting with Exogenous Variables

Xiangfei Qiu [1]   Yuhan Zhu [1]   Zhengyu Li [1]   Xingjian Wu [1]   Bin Yang [1]   Jilin Hu [1 2 3]

## Abstract

Time series forecasting is essential in various domains. Compared to relying solely on endogenous variables (i.e., target variables), considering exogenous variables (i.e., covariates) provides additional predictive information and often leads to more accurate predictions. However, existing methods for *time series forecasting with exogenous variables* (TSF-X) have the following shortcomings: 1) they do not leverage future exogenous variables, 2) they fail to fully account for the correlation between endogenous and exogenous variables. In this study, to better leverage exogenous variables, especially future exogenous variables, we propose **DAG**, which *utilizes Dual correlAtion network along both the temporal and channel dimensions for time series forecasting with exoGenous* variables. Specifically, we propose two core components: the Temporal Correlation Module and the Channel Correlation Module. Both modules consist of a correlation discovery submodule and a correlation injection submodule. The former is designed to capture the correlation effects of historical exogenous variables on future exogenous variables and on historical endogenous variables, respectively. The latter injects the discovered correlation relationships into the processes of forecasting future endogenous variables based on historical endogenous variables and future exogenous variables.

## 1. Introduction

Time series forecasting plays a vital role in many real-world applications, including economic (Wu et al., 2024; Wang

[1]School of Data Science and Engineering, ECNU [2]Shanghai Engineering Research Center of Big Data Management [3]National Institutes of Educational Policy Research, ECNU. Correspondence to: Jilin Hu <jlhu@dase.ecnu.edu.cn>.

*Proceedings of the 43rd International Conference on Machine Learning*, Seoul, South Korea. PMLR 306, 2026. Copyright 2026 by the author(s).

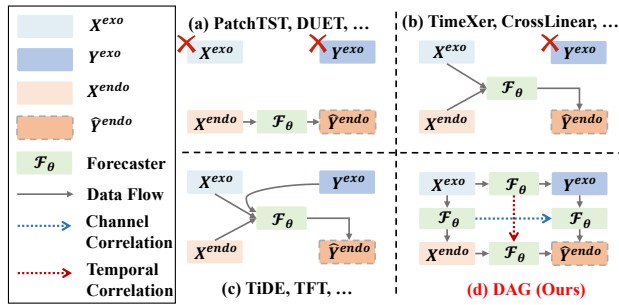

*Figure 1.* Time series forecasting algorithms can be classified as follows: (a) univariate/multivariate algorithms without exogenous variables, e.g., PatchTST (Nie et al., 2023) and DUET (Qiu et al., 2025c); (b) algorithms considering only historical exogenous variables, e.g., TimeXer (Wang et al., 2024) and CrossLinear (Zhou et al., 2025); (c) algorithms that account for both historical and future exogenous variables, e.g., TiDE (Das et al., 2023) and TFT (Lim et al., 2021); and (d) algorithms that consider both exogenous variables and correlation relationships.

et al., 2025; Huang et al., 2022), traffic (Qiu et al., 2026; Xu et al., 2023; Wang et al., 2026b; Wu et al., 2026; Shao et al., 2025), health (Wu et al., 2025a; Wang et al., 2026c; Cheng et al., 2026; Miao et al., 2021), energy (Alvarez et al., 2011; Wang et al., 2026a; Liu et al., 2026a), and AIOps (Wang et al., 2023; Qi et al., 2023; Yu et al., 2025a). Given historical observations, it is valuable if we can know the future values ahead of time (Yu et al., 2025b; Qiu et al., 2025b;a; Liu et al., 2026b; Wu et al., 2025c;b;d). With the rapid development of deep learning, numerous forecasting methods have been proposed, most of which focus on univariate or multivariate time series and rely on learning the temporal dependencies within a single endogenous (i.e., target) variable or across multiple endogenous variables—see Figure 1a.

However, beyond endogenous variables themselves, many practical scenarios involve another type of information that significantly influences the accuracy of predictions—exogenous variables (i.e., covariates). Particularly in scenarios where future exogenous variables are available, effectively leveraging such auxiliary information can substantially enhance forecasting performance. For example, in traffic prediction, having access to future weather conditions can improve the estimation of future traffic volume; in

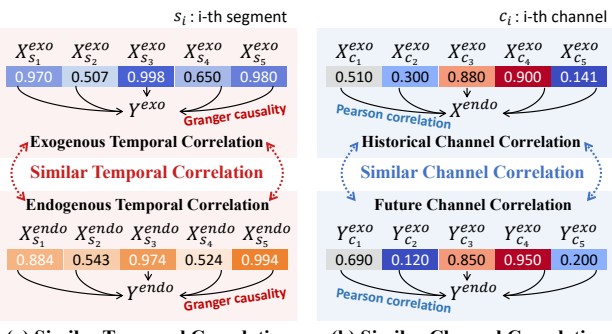

*Figure 2.* Diagram of Temporal Correlation and Channel Correlation. Granger causality is used to measure the correlation between historical data and future data. Pearson correlation is used to measure the correlation between exogenous and endogenous variables.

retail sales forecasting, incorporating holiday schedules and promotional campaigns can lead to better inventory planning. Although these covariates are not part of the prediction target itself, they often contain strong predictive signals.

From a temporal perspective, covariates can be categorized into two types: historical exogenous variables, which are exogenous information observed during the historical period, and future exogenous variables, which are exogenous information known for the forecast horizon. Despite their high predictive value, current deep learning approaches still underutilize future exogenous variables. As shown in Figure 1, existing exogenous-aware forecasting methods can be broadly divided into the following two categories: 1) Methods using only historical information (Figure 1b): These approaches rely solely on historical endogenous and exogenous variables to forecast future endogenous variables. Representative models include TimeXer (Wang et al., 2024) and CrossLinear (Zhou et al., 2025). By entirely ignoring future covariates, these methods may underperform in scenarios where such information is available in advance. 2) Methods that use both historical and future exogenous variables (Figure 1c): For instance, TiDE (Das et al., 2023) and TFT (Lim et al., 2021) use historical information and future exogenous variables to forecast future endogenous variables. However, without fully modeling correlation constraints, such approaches are susceptible to spurious correlations.

Further analysis—see Figure 2, reveals that forecasting with known future covariates involves correlation dependencies across both temporal and channel dimensions. Temporally, the influence of historical exogenous variables on future exogenous variables mirrors the evolution from historical to future endogenous variables. In terms of channels, the interaction patterns between historical exogenous and endogenous variables are often transferable to those between future exogenous and endogenous variables. This dual correlation structure remains an underexplored but critical feature that is insufficiently addressed by existing methods.

Based on the analysis of existing work and our observations in real-world data, we propose a general framework, **DAG**, which *utilizes Dual correlAtion network along both the temporal and channel dimensions for time series forecasting with exoGenous* variables, enabling high-quality predictions of future endogenous variables. Specifically, we first introduce the *Temporal Correlation Module*. Since the influence of historical exogenous variables on future exogenous variables structurally resembles the evolution of historical endogenous variables into future endogenous variables, we design a correlation discovery module to capture how historical exogenous variables affect future exogenous variables. We then construct a correlation injection module that incorporates the discovered correlation relationships into the process of forecasting future endogenous variables based on historical endogenous variables. Next, we propose the *Channel Correlation Module*, which follows a similar design principle, where a correlation discovery module models how historical exogenous variables influence historical endogenous variables, and a correlation injection module incorporates the discovered relationships to enhance the prediction of future endogenous variables based on future exogenous variables. Finally, we combine the temporal correlation loss, channel correlation loss, and the forecasting loss of future endogenous variables as the overall *loss function*, enabling end-to-end optimization of the forecasting task. Our contributions are summarized as:

- We propose a general framework called DAG, which improves forecasting accuracy by discovering and injecting correlation relationships across both temporal and channel dimensions, fully leveraging exogenous variables.

- We design a correlation module to capture historical exogenous impacts on future exogenous and historical endogenous variables.

- We design a correlation injection module that integrates the discovered temporal and channel correlation into the forecasting process of future endogenous variables.

- We open-source our own collected TSF-X datasets and conduct extensive experiments on both public and newly released datasets. The results demonstrate that DAG outperforms state-of-the-art methods. Additionally, all datasets and code are available at: https://github.com/decisionintelligence/DAG.

## 2. Related works

### 2.1. Univariate and Multivariate Forecasting

Existing time series forecasting models are typically classified into univariate and multivariate forecasting methods based on the number of input and output variables. **Univariate time series forecasting models** rely solely on the

historical values of a single variable to predict future values. Traditional univariate forecasting methods, such as ARIMA (Box & Pierce, 1970), ETS (Hyndman et al., 2008), and Theta (Federico Garza, 2022) are classical and widely used techniques. However, these methods still depend on manual feature engineering and model design, which limits their flexibility and automation. With the rapid advancement of deep learning technologies, methods like N-BEATS and DeepAR can automatically learn patterns from historical data, excelling at capturing nonlinear relationships and long-term dependencies. On the other hand, **multivariate time series forecasting models** use multiple input variables to predict the corresponding output variables. The classic methods include VAR (Godahewa et al., 2021), Random Forests(Breiman, 2001) and LightGBM (Ke et al., 2017). In recent years, with the rise of deep learning, various architectural approaches have gained widespread attention. For instance, Transformer architectures, such as Informer (Zhou et al., 2021), FEDformer (Zhou et al., 2022), Autoformer (Wu et al., 2021), Triformer (Cirstea et al., 2022), and PatchTST (Nie et al., 2023), can more accurately capture the complex relationships between temporal tokens. MLP-based methods, such as SparseTSF (Lin et al., 2024a), CycleNet (Lin et al., 2024b), DUET (Qiu et al., 2025c), NLinear (Zeng et al., 2023), and DLinear (Zeng et al., 2023), utilize simpler architectures with fewer parameters but still achieve highly competitive forecasting accuracy. However, all those methods overlook an important practical factor—exogenous (historical or future exogenous). In many real-world scenarios, exogenous data is known or can be approximately known, and utilizing exogenous data can significantly improve the accuracy of predictions.

## 2.2. Forecasting with Exogenous Variables

Time series forecasting with exogenous variables has been extensively discussed in classical statistical methods. Some statistical methods have been extended to incorporate exogenous variables as part of the input. Methods like ARIMAX (Williams, 2001) and SARIMAX (Vagropoulos et al., 2016) have long utilized exogenous variables to enhance forecasting accuracy. More recently, deep learning approaches have advanced this area: CrossLinear (Zhou et al., 2025) uses cross-correlation embeddings to capture dependencies between historical endogenous and exogenous variables; NBEATSx (Olivares et al., 2023) extends N-BEATS with dedicated branches to utilize both past and future exogenous inputs; TiDE (Das et al., 2023) employs an MLP-based architecture to integrate static and future covariates by concatenation with endogenous features at each time step. The Temporal Fusion Transformer (TFT) (Lim et al., 2021) integrates historical and current exogenous variables using attention mechanisms. Furthermore, TimeXer(Wang et al., 2024) introduces patch-wise embeddings to flexibly incorpo-

rate exogenous covariates without strict temporal alignment. ExoTST (Tayal et al., 2024) utilizes innovative embedding, cross-attention, and cross-temporal fusion within an attention framework to robustly handle time lags and missing data. On the other hand, GCGNet (Li et al., 2026c) models correlations using a graph structure, but does not explicitly distinguish past and future variables or endogenous and exogenous variables. However, these methods generally rely on relatively simple combinations of historical inputs and future exogenous, without fully modeling the complex interactions among historical endogenous, historical exogenous, and future exogenous variables.

## 3. Preliminaries

### 3.1. Definitions

**Definition 3.1** (Time Series). A time series $Z \in \mathbb{R}^{N \times T}$ contains $T$ equal-spaced time points with $N$ channels. If $N = 1$, the time series is called univariate; otherwise, it is multivariate when $N > 1$. For clarity, we denote $X_{i,j}$ as the $j$-th time point of the $i$-th channel.

**Definition 3.2** (Endogenous Time Series). An endogenous time series, denoted as $X^{endo} \in \mathbb{R}^{N \times T}$, refers to the primary target variable(s) whose future values we aim to forecast. These variables are determined by internal system dynamics and may depend on their own past as well as other external inputs.

**Definition 3.3** (Exogenous Time Series). An exogenous time series refers to covariate variables that are not the direct forecasting targets but may influence the endogenous time series. These variables are often known or can be estimated in advance, such as weather, calendar events, or holidays. We denote the exogenous variates as two parts: the *past exogenous variables* $X^{\text{exo}} \in \mathbb{R}^{D \times T}$ and the *future exogenous variables* $Y^{\text{exo}} \in \mathbb{R}^{D \times F}$, where $D$ is the number of exogenous variates, $T$ is the number of past time steps, and $F$ is the forecasting horizon.

### 3.2. Problem Statement

**Exogenous-Aware Time Series Forecasting.** Given a historical endogenous time series $X^{\text{endo}} \in \mathbb{R}^{N \times T}$, along with corresponding past exogenous variables $X^{\text{exo}} \in \mathbb{R}^{D \times T}$ and future exogenous variables $Y^{\text{exo}} \in \mathbb{R}^{D \times F}$, the objective is to predict the future values of the endogenous time series over a forecasting horizon of length $F$. Formally, the goal is to design a function $\mathcal{F}_\theta$ parameterized by $\theta$, which models the input time series to get the forecasted future:

$$\hat{Y}^{\text{endo}} = \mathcal{F}_\theta(X^{\text{endo}}, X^{\text{exo}}, Y^{\text{exo}}), \qquad (1)$$

where $\hat{Y}^{\text{endo}} \in \mathbb{R}^{N \times F}$ denotes the predicted future endogenous variables. $X^{\text{endo}}, X^{\text{exo}}, Y^{\text{exo}}$ denote historical endogenous variables, historical exogenous variables, and future

exogenous variables, respectively.

# 4. Methodology

## 4.1. Structure Overview

Figure 3 illustrates the overall architecture of DAG, which effectively improves forecasting accuracy by discovering and injecting correlation relationships along both the temporal and channel dimensions, thereby fully leveraging the information contained in exogenous variables. Specifically, we first introduce the *Temporal Correlation Module*. Since the influence of historical exogenous variables on future exogenous variables structurally resembles the evolution of historical endogenous variables into future endogenous variables, we design a correlation discovery module to capture how historical exogenous variables affect future exogenous variables. We then construct a correlation injection module that incorporates the discovered correlation relationships into the process of forecasting future endogenous variables based on historical endogenous variables. Next, we propose the *Channel Correlation Module*, which follows a similar design principle, where a correlation discovery module models how historical exogenous variables influence historical endogenous variables, and a correlation injection module incorporates the discovered relationships to enhance the prediction of future endogenous variables based on future exogenous variables. Finally, we combine the temporal correlation loss, channel correlation loss, and the forecasting loss of future endogenous variables as the overall *loss function*, enabling end-to-end optimization of the task.

## 4.2. Temporal Correlation Module

As analyzed in Section 1, the influence mechanism of historical exogenous variables on future exogenous variables shares a structural similarity with the evolution process from historical endogenous variables to future endogenous variables. Therefore, the *Temporal Correlation Module* is designed to uncover the correlation between historical and future exogenous variables (*Temporal Correlation Discovery Module*), and to inject the discovered correlation into the modeling process of forecasting future endogenous variables based on historical endogenous variables (*Temporal Correlation Injection Module*), thereby improving accuracy.

### 4.2.1. TEMPORAL CORRELATION DISCOVERY

To extract the correlation between historical and future exogenous variables, the *Temporal Correlation Discovery Module* adopts a patch-wise representation strategy (Nie et al., 2023; Cirstea et al., 2022). Specifically, each historical exogenous variable is segmented into multiple patches,

and each patch is projected into a temporal token.

$$S_i = \text{PatchEmbed}(\text{Patchify}(X_i^{\text{exo}}, i \in \{1, ..., D\})), \quad (2)$$

where $S_i = [s_{i,1}, s_{i,2}, ..., s_{i,M}] \in \mathbb{R}^{M \times d}$ denotes all tokens of the $i$-th channel. $M$ is the number of patches. $s_{i,j} \in \mathbb{R}^d$ denotes the $j$-th token of $i$-th channel. PatchEmbed maps each patch, added by its position embedding, into a d-dimensional vector via a linear projector. We then employ a standard Transformer block to model the influence weights of different patches on the future exogenous variable.

$$Q = S_i \cdot W_{q'}, \; K = S_i \cdot W_{k'}, \; V = S_i \cdot W_{v'}, \quad (3)$$

$$score = \sigma(\frac{Q \cdot K^T}{\sqrt{d}}), S_i' = score \cdot V, \quad (4)$$

where $S_i' \in \mathbb{R}^{M \times d}$ is the representations processed by MSA and $\sigma$ is the softmax operation.

To enhance robustness, instead of directly passing the generated attention score to the Temporal Correlation Injection Module, we extract and transfer the learnable parameters—specifically the query ($W_{q'}$) and key ($W_{k'}$) matrices—of the Multi-Head Self-Attention (MSA) mechanism that generates the attention score. These parameters serve as the temporal correlation representation to be injected.

Finally, the temporal tokens $S_i'$ pass through a FeedForward layer to yield $\hat{S}_i \in \mathbb{R}^{M \times d}$, which is subsequently flattened and projected to forecast future exogenous variables $\hat{Y}^{exo}$.

$$\hat{Y}_i^{exo} = \text{Linear}(\text{Flatten}(\hat{S}_i)). \quad (5)$$

The prediction loss of the future exogenous variables is used as the temporal correlation loss during training:

$$L_t = \left\| Y^{exo} - \hat{Y}^{exo} \right\|_1. \quad (6)$$

### 4.2.2. TEMPORAL CORRELATION INJECTION

Consistent with the approach used in the *Temporal Correlation Discovery Module*, we first obtain patch-wise representations of the historical endogenous variables.

$$P_i = \text{PatchEmbed}(\text{Patchify}(X_i^{\text{endo}}, i \in \{1, ..., N\})), \quad (7)$$

where $P_i = [p_{i,1}, p_{i,2}, ..., p_{i,M}] \in \mathbb{R}^{M \times d}$ denotes all tokens of the $i$-th channel. $p_{i,j} \in \mathbb{R}^d$ denotes the $j$-th token of $i$-th channel.

Then, we employ the Correlation Transformer Block to model the influence of different endogenous variable patches on the future endogenous variable, while incorporating the correlation information passed from the *Temporal Correlation Discovery Module*. Specifically, in the attention mechanism, the input tokens are projected via original

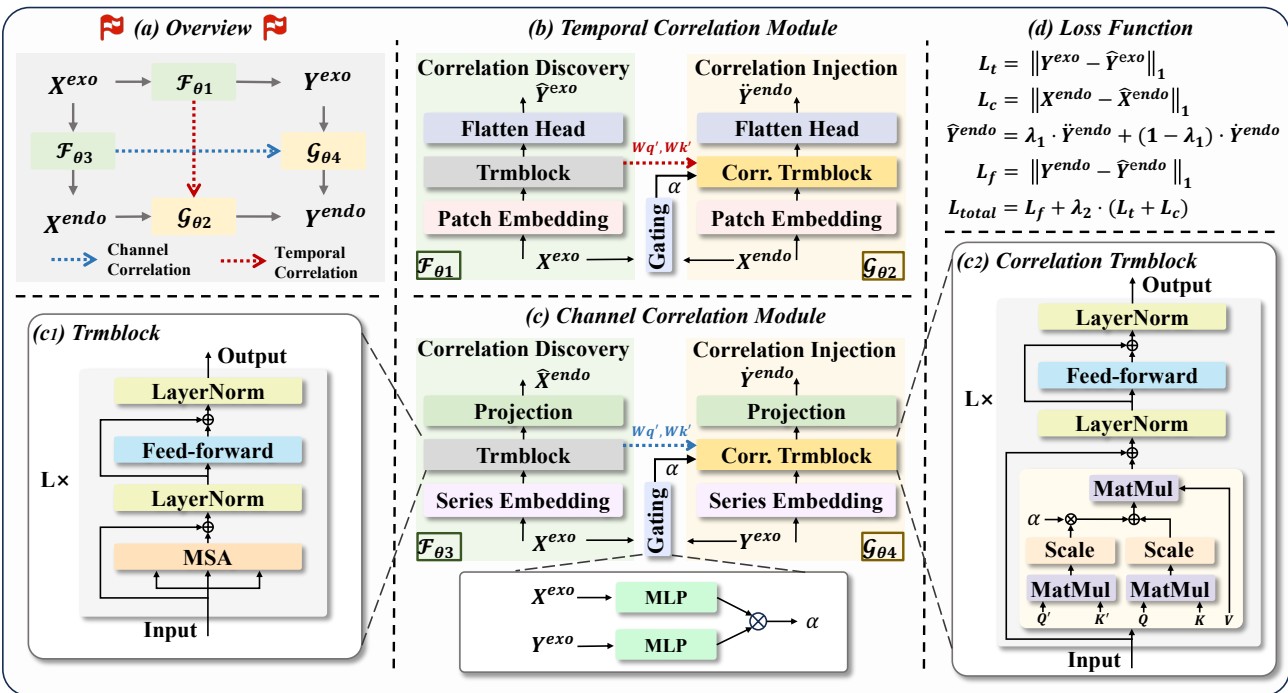

*Figure 3.* The architecture of DAG. (a) Overview of the DAG framework, which comprises Temporal Correlation Modules ($\mathcal{F}_{\theta_1}$ and $\mathcal{G}_{\theta_2}$) and Channel Correlation Modules ($\mathcal{F}_{\theta_3}$ and $\mathcal{G}_{\theta_4}$). (b) Detailed structure of the Temporal Correlation Module. (c) Detailed structure of the Channel Correlation Module. (c1) The standard Transformer block (Trmblock). (c2) The Correlation Trmblock, which injects learned correlation into the Trmblock. (d) The loss function. Note that the Gating, Trmblock, and Correlation Trmblock used in (b) and (c) share the same architecture.

query and key matrices ($W_q$, $W_k$ and $W_v$) within the Correlation Trmblock to obtain $Q$, $K$ and $V$, and simultaneously through ($W_{q'}$ and $W_{k'}$) extracted from the Temporal Correlation Discovery Module to generate $Q'$ and $K'$.

$$Q' = P_i \cdot W_{q'}, \; K' = P_i \cdot W_{k'}, \qquad (8)$$
$$Q = P_i \cdot W_q, \; K = P_i \cdot W_k, \; V = P_i \cdot W_v. \qquad (9)$$

We then compute two sets of attention scores, followed by scaling and softmax operations ($\sigma$). A learnable weighting factor $\alpha$ is then introduced to fuse the two attention scores, resulting in the final fused attention score. This mechanism explicitly injects temporal correlation structure into the attention computation, enabling the model to capture Correlation informed dependencies among tokens along the temporal dimension.

$$S_{\text{fused}} = \alpha \cdot \sigma(\frac{Q \cdot K^T}{\sqrt{d}}) + (1 - \alpha) \cdot \sigma(\frac{Q' \cdot K'^T}{\sqrt{d}}), \quad (10)$$
$$P_i' = \sigma(S_{\text{fused}}) \cdot V. \qquad (11)$$

**Gating Mechanism:** To adaptively weight the two sets of attention scores along the temporal dimension, we design a Gating mechanism—see Equation 12. This module takes the historical exogenous variables $X^{exo}$ and the historical endogenous variables $X^{endo}$ as input, and encodes them

through two separate multilayer perceptrons (MLPs) to obtain nonlinear representations. The outputs of the two MLPs are then fused via a dot product operation to produce a scalar weight $\alpha$. This weight $\alpha$ is used to adaptively combine the attention scores, guiding the model to integrate information dynamically based on the actual influence strength between historical exogenous and endogenous variable, thereby improving modeling accuracy and robustness.

$$\alpha = \text{MLP}(X_i^{exo})^{\top} \cdot \text{MLP}(X_i^{endo}). \qquad (12)$$

Finally, the temporal tokens $P_i'$ output by the Attention mechanism are then processed by Feedforward layer to obtain $\hat{P}_i \in \mathbb{R}^{F \times d}$ and fed into the flatten prediction head to generate forecasts of future endogenous variables.

$$\ddot{Y}^{endo} = \text{Linear}(\text{Flatten}(\hat{P}_i)). \qquad (13)$$

### 4.3. Channel Correlation Module

Similar to the temporal dimension, the *Channel Correlation Module* aims to model and inject correlation along the variable (channel) dimension. This module focuses on how historical exogenous variables influence historical endogenous variables and how such correlation patterns can be transferred to enhance the prediction of future endogenous variables using future exogenous variables.

### 4.3.1. CHANNEL CORRELATION DISCOVERY

To capture the correlation effects from historical exogenous variables to historical endogenous variables, the *Channel Correlation Discovery Module* employs a series-wise representation. Specifically, we encode each historical exogenous variables $X_i^{exo}$ over time into a series-wise token through a series embedding layer.

$$U = \text{SeriesEmbedding}(X_i^{\text{exo}}, i \in \{1, ..., D\}), \quad (14)$$

The embedded historical exogenous variables $U = [u_1, u2, \cdots, u_D] \in \mathbb{R}^{D \times d}$ are then fed into a standard Transformer block (Trmblock):

$$\mathcal{Q} = U \cdot \mathcal{W}_{q'}, \ \mathcal{K} = U \cdot \mathcal{W}_{k'}, \ \mathcal{V} = U \cdot \mathcal{W}_{v'}, \quad (15)$$

$$U' = \sigma(\frac{\mathcal{Q} \cdot \mathcal{K}^T}{\sqrt{d}}) \cdot \mathcal{V}, \quad (16)$$

then the series token $U'$ is processed through FeedForward layer and obtain $\hat{U} \in \mathbb{R}^{D \times d}$ and passed through a projection layer to generate a prediction $\hat{X}^{endo}$ for the historical endogenous variables:

$$\hat{X}^{endo} = \text{Linear}(\text{MLP}(\hat{U})). \quad (17)$$

Similar to Section 4.2.1, the Channel Correlation Discovery Module serves two purposes: (1) it is used to define the channel correlation loss, and (2) its underlying attention mechanism's learnable parameters (query and key matrices $\mathcal{W}_{q'}$ and $\mathcal{W}_{k'}$) are extracted to be injected into the following channel correlation injection module.

The prediction loss of the historical endogenous variables is used as the channel correlation loss during training:

$$L_c = \left\| X^{endo} - \hat{X}^{endo} \right\|_1. \quad (18)$$

### 4.3.2. CHANNEL CORRELATION INJECTION

This module uses future exogenous variables to predict future endogenous variables, while incorporating the channel-level correlation representations extracted in the previous step. First, the future exogenous variables are encoded using the same series embedding strategy:

$$O = \text{SeriesEmbedding}(Y_i^{\text{exo}}, i \in \{1, ..., D\}), \quad (19)$$

The Correlation Trmblock is then used to model channel dependencies while injecting learned correlation information.

$$\mathcal{Q}' = O \cdot \mathcal{W}_{q'}, \mathcal{K}' = O \cdot \mathcal{W}_{k'}, \quad (20)$$

$$\mathcal{Q} = O \cdot \mathcal{W}_q, \mathcal{K} = O \cdot \mathcal{W}_k, \mathcal{V} = O \cdot \mathcal{W}_v, \quad (21)$$

$$\alpha = \text{MLP}(X^{exo})^T \cdot \text{MLP}(Y^{exo}), \quad (22)$$

$$S_{\text{fused}} = \alpha \cdot \sigma(\frac{\mathcal{Q} \cdot \mathcal{K}^T}{\sqrt{d}}) + (1 - \alpha) \cdot \sigma(\frac{\mathcal{Q}' \cdot \mathcal{K}'^T}{\sqrt{d}}), \quad (23)$$

$$O' = \sigma(S_{\text{fused}}) \cdot \mathcal{V}. \quad (24)$$

**Gating Mechanism:** Similar to the gating mechanism in Section 4.2.2. This gating mechanism takes the historical exogenous variable $X^{exo}$ and future exogenous variable $Y^{exo}$ as input, and encodes them using two separate MLPs. The outputs of the two MLPs are then fused via a dot product operation to compute a scalar weight $\alpha$. This weight $\alpha$ is subsequently used to adaptively combine the attention scores, dynamically adjusting the model's focus on the historical and future exogenous information based on their relative influence strength.

The final fused attention is then used to refine the representation of the series token. The refined tokens are processed through FeedForward layer to obtain $\hat{O}$ and then passed into the prediction head to generate a prediction $\dot{Y}^{endo}$ for the future endogenous variables:

$$\dot{Y}^{endo} = \text{Linear}(\text{MLP}(\hat{O})). \quad (25)$$

### 4.4. Loss Function

The training objective of DAG consists of three components: 1) the *temporal correlation loss* $L_t$ introduced in Section 4.2.1, which measures how well the model captures temporal correlation structures when forecasting future exogenous variables. 2) the *channel correlation loss* $L_c$ from Section 4.3.1, which reflects the modeling error in capturing the correlation relationships between historical exogenous and endogenous variables. 3) the final *forecasting loss* $L_f$.

The final prediction is derived by fusing two candidates with a fusion weight $\lambda_1$:

$$\hat{Y}^{endo} = \lambda_1 \cdot \ddot{Y}^{endo} + (1 - \lambda_1) \cdot \dot{Y}^{endo}, \quad (26)$$

where $\ddot{Y}^{endo}$ and $\dot{Y}^{endo}$ are prediction outputs from Temporal Correlation Injection Module and Channel Correlation Injection Module. The forecasting loss is defined as:

$$L_f = \left\| Y^{endo} - \hat{Y}^{endo} \right\|_1. \quad (27)$$

The total loss combines forecasting and correlation losses:

$$L_{total} = L_f + \lambda_2 \cdot (L_t + L_c), \quad (28)$$

where $\lambda_2$ is a correlation weight that balances the contribution of correlation modeling during training.

## 5. Experiments

In Section 5.1, we introduce the datasets, baselines, and implementation details. Section 5.2 presents the main experimental results, while Section 5.3 provides detailed analyses, including experiments without future exogenous variables 5.3.1, parameter sensitivity studies 5.3.2, ablation studies 5.3.3, and extended lookback experiments 5.3.4. Due to space limitations, visualization experiments and full results are provided in Appendices C.1 and C.2, respectively.

*Table 1.* Statistics of datasets. Ex. and En. are abbreviations for the Exogenous variable and Endogenous variable, respectively.

| Dataset | #Num | Ex. Descriptions | En. Descriptions | Sampling Frequency | Lengths | Split |
|---|---|---|---|---|---|---|
| ETTh | 6 | Power Load Feature | Oil Temperature | 1 Hour | 14,400 | 6:2:2 |
| ETTm | 6 | Power Load Feature | Oil Temperature | 15 Minutes | 57,600 | 6:2:2 |
| Weather | 20 | Climate Feature | CO2-Concentration | 10 Minutes | 52,696 | 7:1:2 |
| Exhange | 7 | Exchange Rate | Exchange Rate | 1 Day | 7,588 | 7:1:2 |
| Electricity | 320 | Electricity Consumption | Electricity Consumption | 1 Hour | 26,304 | 7:1:2 |
| Traffic | 861 | Road Occupancy Rates | Road Occupancy Rates | 1 Hour | 17,544 | 7:1:2 |
| NP | 2 | Grid Load, Wind Power | Nord Pool Electricity Price | 1 Hour | 52,416 | 7:1:2 |
| PJM | 2 | System Load, SyZonal COMED load | Pennsylvania-New Jersey-MarylandElectricity Price | 1 Hour | 52416 | 7:1:2 |
| BE | 2 | Generation, System Load | Belgium's Electricity Price | 1 Hour | 52,416 | 7:1:2 |
| FR | 2 | Generation, System Load | France's Electricity Price | 1 Hour | 52,416 | 7:1:2 |
| DE | 2 | Wind power, Ampirion zonal load | German's Electricity Price | 1 Hour | 52,416 | 7:1:2 |
| Energy | 5 | Battery, Geothermal, Hydroelectric, Solar, Wind | Thermoelectric | 1 Hour | 13,064 | 7:1:2 |
| Colbún | 2 | Precipitation, Tributary Inflow | Water Level | 1 Day | 2,958 | 7:1:2 |
| Rapel | 2 | Precipitation, Tributary Inflow | Water Level | 1 Day | 3,366 | 7:1:2 |
| Sdwpfh | 6 | Climate Feature | Active Power | 1 Hour | 1,4641 | 7:1:2 |
| Sdwpfm | 6 | Climate Feature | Active Power | 30 Minutes | 2,9281 | 7:1:2 |

*Table 2.* Average results on 12 real-world datasets that satisfy the TSF-X task, where the inputs are ($X^{\text{endo}}$, $X^{\text{exo}}$, and $Y^{\text{exo}}$). **Red**: the best, Blue: the 2nd best. Full results are available in Table 5 in Appendix C.2.

| Models | DAG (ours) | | GCGNet | | TimeXer | | TFT | | TiDE | | DUET | | CrossLinear | | Amplifier | | TimeKAN | | PatchTST | |
|---|---|---|---|---|---|---|---|---|---|---|---|---|---|---|---|---|---|---|---|---|---|
| Metrics | mse | mae | mse | mae | mse | mae | mse | mae | mse | mae | mse | mae | mse | mae | mse | mae | mse | mae | mse | mae |
| NP | **0.362** | **0.344** | 0.370 | 0.348 | 0.418 | 0.371 | 0.379 | 0.375 | 0.443 | 0.400 | 0.411 | 0.408 | 0.371 | 0.387 | 0.420 | 0.418 | 0.405 | 0.419 | 0.390 | 0.396 |
| PJM | **0.093** | **0.180** | 0.095 | 0.187 | 0.108 | 0.198 | 0.114 | 0.207 | 0.142 | 0.246 | 0.102 | 0.197 | 0.112 | 0.223 | 0.137 | 0.246 | 0.139 | 0.262 | 0.133 | 0.259 |
| BE | **0.423** | **0.279** | 0.431 | 0.294 | 0.452 | 0.290 | 0.454 | 0.291 | 0.498 | 0.325 | 0.515 | 0.354 | 0.479 | 0.337 | 0.559 | 0.413 | 0.548 | 0.407 | 0.577 | 0.432 |
| FR | **0.414** | **0.219** | 0.415 | 0.234 | 0.427 | 0.241 | 0.504 | 0.257 | 0.484 | 0.281 | 0.496 | 0.327 | 0.483 | 0.298 | 0.554 | 0.408 | 0.547 | 0.374 | 0.588 | 0.410 |
| DE | **0.370** | **0.370** | 0.401 | 0.389 | 0.475 | 0.418 | 0.489 | 0.446 | 0.499 | 0.447 | 0.482 | 0.430 | 0.485 | 0.452 | 0.473 | 0.441 | 0.473 | 0.445 | 0.501 | 0.455 |
| Energy | **0.124** | **0.267** | 0.131 | 0.277 | 0.163 | 0.315 | 0.130 | 0.283 | 0.153 | 0.302 | 0.203 | 0.367 | 0.239 | 0.402 | 0.233 | 0.389 | 0.218 | 0.381 | 0.226 | 0.377 |
| Sdwpfm1 | **0.423** | 0.461 | 0.424 | **0.457** | 0.701 | 0.609 | 0.482 | 0.474 | 0.483 | 0.507 | 0.599 | 0.570 | 0.426 | 0.502 | 0.437 | 0.490 | 0.447 | 0.534 | 0.435 | 0.502 |
| Sdwpfm2 | 0.477 | **0.485** | 0.475 | 0.486 | 0.803 | 0.653 | 0.476 | 0.488 | 0.486 | 0.516 | 0.514 | 0.490 | 0.533 | 0.573 | 0.491 | 0.512 | 0.497 | 0.564 | 0.510 | 0.547 |
| Sdwpfh1 | **0.448** | **0.486** | 0.450 | 0.500 | 0.746 | 0.643 | 0.479 | 0.491 | 0.453 | 0.508 | 0.539 | 0.516 | 0.557 | 0.593 | 0.537 | 0.598 | 0.577 | 0.638 | 0.468 | 0.527 |
| Sdwpfh2 | 0.523 | 0.530 | 0.520 | 0.536 | 0.891 | 0.719 | 0.566 | **0.521** | 0.599 | 0.583 | 0.647 | 0.566 | 0.538 | 0.574 | 0.521 | 0.581 | 0.647 | 0.672 | 0.549 | 0.580 |
| Colbun | **0.098** | **0.154** | 0.107 | 0.175 | 0.145 | 0.235 | 0.238 | 0.297 | 0.164 | 0.227 | 0.198 | 0.266 | 0.126 | 0.195 | 0.173 | 0.246 | 0.128 | 0.175 | 0.239 | 0.309 |
| Rapel | **0.230** | **0.305** | 0.306 | 0.307 | 0.344 | 0.362 | 0.305 | 0.333 | 0.320 | 0.351 | 0.269 | 0.326 | 0.252 | 0.313 | 0.257 | 0.321 | 0.249 | 0.311 | 0.269 | 0.332 |
| 1st Count | **10** | **10** | 2 | 1 | 0 | 0 | 0 | 1 | 0 | 0 | 0 | 0 | 0 | 0 | 0 | 0 | 0 | 0 | 0 | 0 |

## 5.1. Experimental Settings

### 5.1.1. DATASETS

To ensure comprehensive and fair comparisons across different models, 1) we conduct experiments on 12 real-world datasets that satisfy the TSF-X conditions, including 5 EPF (Lago et al., 2021; Wang et al., 2024) datasets and 7 datasets we sourced ourselves. The future exogenous variables in these datasets are either approximately known or highly accurate. 2) Following TimeXer (Wang et al., 2024) and CrossLinear (Zhou et al., 2025), we also perform experiments on eight common multivariate forecasting datasets (ETT with 4 subsets, Weather, Exchange, Electricity, and Traffic). Notably, although the future exogenous variables in these datasets are not known, we conduct vanilla forecasting with exogenous variables by treating the last dimension of the multivariate data as the endogenous series and the remaining dimensions as exogenous variables. Due to space limitations, the experimental results for this part are provided in Tables 7 and 8 of the appendix. More details about

the datasets are provided in Table 1.

### 5.1.2. BASELINES

We comprehensively compare our model against 9 baselines, including 1) methods that inherently support future exogenous variables, such as GCGNet (Li et al., 2026c), TimeXer (Wang et al., 2024), TFT (Lim et al., 2021), and TiDE (Das et al., 2023), 2) as well as methods that do not originally support future exogenous variables, like DUET (Qiu et al., 2025c), CrossLinear (Zhou et al., 2025), Amplifier (Fei et al., 2025), TimeKAN (Huang et al., 2025), and PatchTST (Nie et al., 2023). For methods that do not support future exogenous variables, we adapt them to incorporate future exogenous variables using an MLP fusion approach—see Algorithm 1 in Appendix A.3.

### 5.1.3. IMPLEMENTATION DETAILS

1) To keep consistent with previous works, we adopt Mean Squared Error (mse) and Mean Absolute Error (mae) as

*Table 3.* Average results under the setting where only historical exogenous variables are used, without future exogenous variables. The inputs are ($X^{\text{endo}}$ and $X^{\text{exo}}$). **Red**: the best, Blue: the 2nd best. Full results are available in Table 6 in Appendix C.2.

| Models | DAG | | GCGNet | | TimeXer | | TFT | | TiDE | | DUET | | CrossLinear | | Amplifier | | TimeKAN | | PatchTST | |
| --- | --- | --- | --- | --- | --- | --- | --- | --- | --- | --- | --- | --- | --- | --- | --- | --- | --- | --- | --- | --- |
| Metrics | mse | mae | mse | mae | mse | mae | mse | mae | mse | mae | mse | mae | mse | mae | mse | mae | mse | mae | mse | mae |
| NP | **0.419** | **0.380** | 0.430 | 0.381 | 0.440 | 0.383 | 0.647 | 0.488 | 0.467 | 0.416 | 0.444 | 0.383 | 0.451 | 0.394 | 0.520 | 0.448 | 0.484 | 0.435 | 0.457 | 0.401 |
| PJM | **0.126** | **0.210** | 0.131 | 0.216 | 0.141 | 0.229 | 0.200 | 0.270 | 0.158 | 0.253 | 0.140 | 0.226 | 0.147 | 0.241 | 0.152 | 0.245 | 0.175 | 0.270 | 0.148 | 0.243 |
| BE | 0.462 | **0.297** | **0.459** | 0.306 | 0.477 | 0.301 | 0.563 | 0.354 | 0.547 | 0.348 | 0.473 | 0.305 | 0.477 | 0.300 | 0.502 | 0.325 | 0.488 | 0.315 | 0.485 | 0.309 |
| FR | **0.435** | 0.249 | 0.435 | **0.246** | 0.454 | 0.247 | 0.535 | 0.288 | 0.494 | 0.290 | 0.468 | 0.262 | 0.476 | 0.257 | 0.494 | 0.286 | 0.491 | 0.276 | 0.470 | 0.264 |
| DE | **0.603** | **0.487** | 0.614 | 0.501 | 0.659 | 0.507 | 0.684 | 0.515 | 0.644 | 0.519 | 0.660 | 0.513 | 0.635 | 0.508 | 0.712 | 0.548 | 0.669 | 0.526 | 0.696 | 0.527 |
| Energy | 0.150 | 0.300 | 0.154 | 0.305 | 0.172 | 0.326 | 0.376 | 0.480 | 0.162 | 0.311 | 0.160 | 0.308 | 0.201 | 0.336 | 0.180 | 0.327 | **0.147** | **0.296** | 0.203 | 0.341 |
| Sdwpfm1 | **0.689** | 0.604 | 0.711 | 0.595 | 0.701 | 0.609 | 0.984 | 0.728 | 0.713 | 0.633 | 0.724 | **0.583** | 0.809 | 0.694 | 0.843 | 0.685 | 0.725 | 0.672 | 0.733 | 0.651 |
| Sdwpfm2 | **0.786** | 0.657 | 0.807 | 0.643 | 0.803 | 0.653 | 1.044 | 0.767 | 0.816 | 0.682 | 0.820 | **0.641** | 0.800 | 0.687 | 0.942 | 0.732 | 0.836 | 0.701 | 0.834 | 0.714 |
| Sdwpfh1 | **0.733** | 0.643 | 0.741 | **0.628** | 0.746 | 0.643 | 0.793 | 0.698 | 0.808 | 0.699 | 0.779 | 0.644 | 0.768 | 0.702 | 1.100 | 0.820 | 0.804 | 0.720 | 0.804 | 0.717 |
| Sdwpfh2 | **0.870** | **0.695** | 0.886 | 0.701 | 0.891 | 0.719 | 0.926 | 0.746 | 0.919 | 0.751 | 1.007 | 0.715 | 0.956 | 0.774 | 0.980 | 0.768 | 0.941 | 0.782 | 1.025 | 0.802 |
| Colbun | **0.117** | **0.155** | 0.119 | 0.165 | 0.132 | 0.219 | 0.556 | 0.386 | 0.188 | 0.240 | 0.147 | 0.198 | 0.129 | 0.195 | 0.152 | 0.210 | 0.201 | 0.253 | 0.150 | 0.221 |
| Rapel | 0.261 | **0.281** | 0.265 | 0.284 | 0.344 | 0.362 | 0.308 | 0.334 | 0.323 | 0.357 | 0.304 | 0.306 | **0.240** | 0.299 | 0.337 | 0.348 | 0.342 | 0.337 | 0.325 | 0.334 |
| 1st Count | **9** | **7** | 1 | 2 | 0 | 1 | 0 | 0 | 0 | 0 | 0 | 2 | 1 | 0 | 0 | 0 | 1 | 1 | 0 | 0 |

evaluation metrics. 2) We conduct both long-term and short-term prediction experiments. For the Colbun and Raperl datasets, the short-term prediction uses a lookback window of 60 with a prediction horizon of 10, while the long-term prediction uses a lookback window of 180 with a prediction horizon of 30. For the other datasets, the short-term prediction uses a lookback window of 168 with a prediction horizon of 24, and the long-term prediction uses a lookback window of 720 with a prediction horizon of 360. 3) We utilize the TFB (Qiu et al., 2024) code repository for unified evaluation, with all baseline results also derived from TFB. Following the settings in TFB (Qiu et al., 2024) and TSFM-Bench (Li et al., 2025), we do not apply the "Drop Last" trick to ensure a fair comparison. Further implementation details can be found in the Appendix A.2.

## 5.2. Main Results

Comprehensive forecasting results are presented in Table 2 to demonstrate the performance of DAG on the TSF-X task. We have the following observations: 1) Compared with forecasters of various structures, DAG achieves outstanding predictive performance. In terms of absolute metrics, Table 2 shows that DAG achieves the highest number of first-place rankings 10 for MSE and 11 for MAE demonstrating clear superiority over other methods such as TimeXer and TFT. 2) Methods that natively support future exogenous variables (e.g., TFT, TimeXer) generally perform better than those that do not (e.g., PatchTST, DLinear), even when enhanced via MLP fusion. DAG, by explicitly modeling correlation relationships while leveraging exogenous inputs, not only competes with these top baselines but surpasses them in nearly all metrics. This demonstrates the strength of incorporating correlation-aware mechanisms in TSF-X tasks.

## 5.3. Model Analyses

### 5.3.1. NOT USING FUTURE EXOGENOUS VARIABLES

Considering that not all datasets have access to future exogenous variables, we conduct experiments using only historical exogenous variables, excluding future exogenous variables, to validate the practicality of DAG—see Table 3. Specifically, for DAG, during inference, we use the $\hat{Y}^{\text{exo}}$ predicted by $\mathcal{F}_{\theta_1}$ to replace the actual value $Y^{\text{exo}}$ used in $\mathcal{G}_{\theta_4}$ for predicting $Y^{\text{endo}}$, thus avoiding the issue of unavailable future exogenous variables. For other baseline models, we also uniformly use only historical exogenous variables. The experimental results show that DAG continues to perform excellently. Methods that consider historical exogenous variable modeling, such as TimeXer and CrossLinear, also perform well. DUET, which flexibly models channel relationships, also achieves good results, while the channel-independent algorithm PatchTST performs poorly.

### 5.3.2. PARAMETER SENSITIVITY

We also conduct parameter sensitivity studies of DAG, with the following observations: 1) Figure 4a and Figure 4b illustrate the performance variations of the DAG model under different settings of the fusion weight $\lambda_1$ and the correlation weight $\lambda_2$. Overall, the model remains stable across a wide range of values, with the optimal range for both $\lambda_1$ and $\lambda_2$ typically falling between 0.3 and 0.7, indicating that moderate weighting is beneficial for improving performance. 2) Figure 4c shows that model dimension has a certain impact on performance, but DAG maintains good stability across different configurations. Smaller dimensions help reduce computational cost, while dimensions in the range of 64 to 256 generally achieve a good balance between prediction accuracy and model complexity. 3) Figure 4d explores the effect of patch length on model performance. Due to differ-

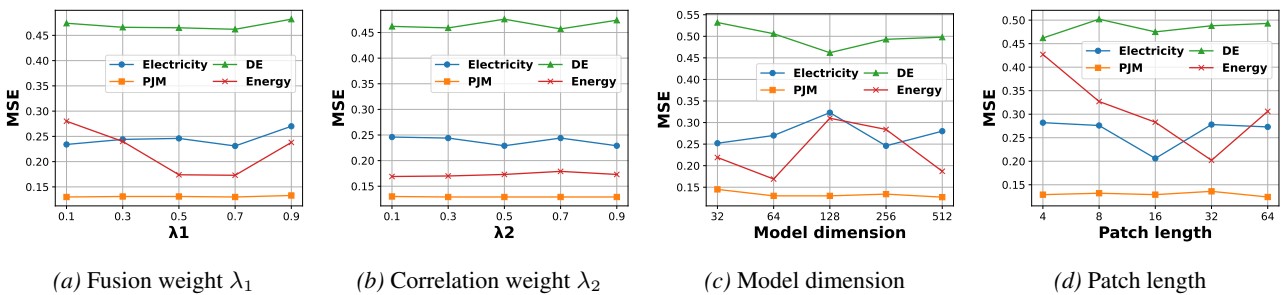

*(a)* Fusion weight $\lambda_1$     *(b)* Correlation weight $\lambda_2$     *(c)* Model dimension     *(d)* Patch length

*Figure 4.* Parameter sensitivity studies of main hyper-parameters in DAG.

*Table 4.* Ablation studies for DAG.

| Datasets | Electricity | | PJM | | DE | | Energy | | Average | |
|---|---|---|---|---|---|---|---|---|---|---|
| Metrics | mse | mae | mse | mae | mse | mae | mse | mae | mse | mae |
| (a) $\mathcal{G}_{\theta_2}$ | 0.512 | 0.514 | 0.191 | 0.264 | 0.846 | 0.586 | 0.263 | 0.408 | 0.453 | 0.443 |
| (b) $\mathcal{G}_{\theta_4}$ | 0.488 | 0.510 | 0.169 | 0.254 | 0.483 | 0.443 | 0.637 | 0.661 | 0.444 | 0.467 |
| (c) $\mathcal{G}_{\theta_2} + \mathcal{G}_{\theta_4}$ | 0.313 | 0.412 | 0.131 | 0.219 | 0.480 | 0.429 | 0.169 | 0.320 | 0.273 | 0.345 |
| (d) $\mathcal{F}_{\theta_1} + \mathcal{G}_{\theta_2}$ + Temporal Correlation | 0.504 | 0.509 | 0.189 | 0.261 | 0.855 | 0.585 | 0.240 | 0.389 | 0.447 | 0.436 |
| (e) $\mathcal{F}_{\theta_3} + \mathcal{G}_{\theta_4}$ + Channel Correlation | 0.392 | 0.459 | 0.170 | 0.255 | 0.473 | 0.438 | 0.350 | 0.467 | 0.346 | 0.405 |
| **DAG (ours)** | **0.246** | **0.370** | **0.130** | **0.218** | **0.462** | **0.418** | **0.169** | **0.320** | **0.252** | **0.331** |

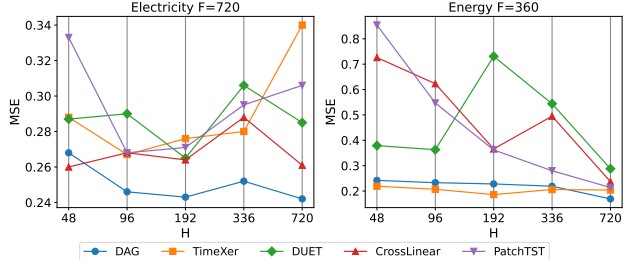

*Figure 5.* Forecasting performance (MSE) with varying look-back windows on 2 datasets: Electricity and Energy. The look-back windows are H = 48, 96, 192, 336 and 720, and the forecasting horizons are F = 720 and 360.

ences in temporal dependencies across datasets, the optimal patch length varies; however, a patch size between 8 and 32 tends to yield favorable results. It is worth noting that very small patch lengths may lead to increased computational complexity, while excessively large patch lengths may weaken the model's ability to capture local dependencies.

### 5.3.3. ABLATION STUDIES

We compare the full version of DAG with the following variants. The Electricity has a lookback length of 96 and forecasts 720 steps, while the other three datasets have a lookback length of 720 steps and forecast 360 steps—see Table 4. We make the following observations: 1) using only historical endogenous variables $\mathcal{G}_{\theta_2}$ or only future exogenous variables $\mathcal{G}_{\theta_4}$ results in suboptimal forecasting performance, indicating that relying on a single input source is insufficient for high-quality predictions. 2) Combining both $\mathcal{G}_{\theta_2} + \mathcal{G}_{\theta_4}$ leads to a significant performance improvement, highlighting their complementary roles in modeling. 3) Furthermore, introducing temporal correlation module ($\mathcal{F}_{\theta_1} + \mathcal{G}_{\theta_2}$ + Temporal Correlation) improves performance compared to using only historical endogenous variables $\mathcal{G}_{\theta_2}$; similarly, incorporating channel correlation module ($\mathcal{F}_{\theta_3}$ + $\mathcal{G}_{\theta_4}$ + Channel Correlation) outperforms the variant that uses only future exogenous variables $\mathcal{G}_{\theta_4}$. These results validate the effectiveness of both correlation modeling mechanisms in DAG. 4) Ultimately, the full DAG model, which integrates both temporal and channel correlation, achieves the best results, demonstrating the importance of modeling dual correlation structures for TSF-X tasks.

### 5.3.4. INCREASING LOOK-BACK WINDOWS

In time series forecasting tasks, the size of the look-back window determines how much historical information the model receives. We select models with better predictive performance from the main experiments as baselines. We configure different look-back windows to evaluate the effectiveness of DAG and visualize the prediction results for look-back window H of 48, 96, 192, 336, 720, and the forecasting horizons are F = 720, 360. From Figure 5, DAG consistently outperforms the baselines on the Electricity, and Energy. As the look-back window increases, the prediction metrics of DAG continue to decrease, indicating that it is capable of modeling longer sequences.

## 6. Conclusion

This study proposes **DAG**, which *utilizes Dual correlAtion network along both the temporal and channel dimensions for time series forecasting with exoGenous* variables, especially by leveraging future exogenous variable information. Specifically, DAG introduces a *Temporal Correlation Module* to capture the influence of historical exogenous variables on future exogenous variables and inject these correlations into endogenous forecasting. In addition, a *Channel Correlation Module* models the relationships between historical exogenous and endogenous variables to further enhance forecasting with future exogenous information.

## Acknowledgements

This work was partially supported by National Natural Science Foundation of China (62472174), the ECNU Multifunctional Platform for Innovation (001), and the Fundamental Research Funds for the Central Universities (YBNLTS2026010). Jilin Hu is the corresponding author of the work.

## Impact Statement

This paper presents work whose goal is to advance the field of Machine Learning. There are many potential societal consequences of our work, none which we feel must be specifically highlighted here.

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

# A. Experimental Details

## A.1. Datasets

We perform experiments on eight common multivariate forecasting datasets: (1) **ETT** includes four subsets: ETTh1 and ETTh2 provide hourly recordings, while ETTm1 and ETTm2 contain data collected every 15 minutes. The endogenous variable is oil temperature, and the six power load features are used as exogenous variables. (2) **Weather** consists of 21 meteorological variables recorded every 10 minutes during 2020 at the Weather Station of the Max Planck Biogeochemistry Institute. The endogenous variable is the Wet Bulb factor, while the remaining indicators are used as exogenous variables. (3) **Electricity** consists of hourly electricity consumption data from 321 clients. The electricity consumption of the last client is the endogenous variable, and the consumption data from the other clients are used as exogenous variables. (4) **Exchange** collects daily exchange rates of eight countries from 1990 to 2016. The exchange rate of the last country is the endogenous variable, and the exchange rates of the others serve as exogenous variables. (5) **Traffic** records hourly road occupancy rates from 862 sensors on Bay Area freeways. We use the last sensor as the endogenous variable and others as exogenous variables.

we conduct experiments on 12 real-world datasets that satisfy the TSF-X conditions, including 5 EPF (Lago et al., 2021; Wang et al., 2024) datasets and 7 datasets we sourced ourselves: (1) **NP** records Nord Pool market with hourly electricity price as the endogenous variable, and grid load and wind power forecast as exogenous features. (2) **PJM** records the Pennsylvania–New Jersey–Maryland Interconnection market, with zonal electricity price in the Commonwealth Edison (COMED) area as the endogenous variable, and corresponding system load and COMED load forecasts as exogenous variables. (3) **BE** records Belgium's electricity market, with hourly electricity price as the endogenous variable, and load forecast in Belgium together with generation forecast from France as exogenous variables. (4) **FR** records the electricity market in France, where the electricity price is the endogenous variable, and generation and load forecasts are used as exogenous variables. (5) **DE** records the German electricity market, with hourly electricity price as the endogenous variable, and zonal load forecast in the Amprion area along with wind and solar generation forecasts as exogenous variables.

In addition to the 5 EPF (Lago et al., 2021; Wang et al., 2024) datasets, we also conduct experiments on 7 datasets that we sourced ourselves, which satisfy the TSF-X conditions. (1) **Energy** provides hourly power generation data collected from the national grid operator in Chile, encompassing multiple energy sources such as battery storage, wind, hydro, solar, and thermoelectric generation. The thermo-

---

**Algorithm 1** MLP Fusion Approach

**Input:** model parameters $\theta_{\text{Model}}$, MLP parameters $\theta_{\text{MLP}}$; learning rate $\eta$; iterations $N$; training dataset $D$; historical endogenous variables $X^{\text{endo}}$; historical exogenous variables $X^{\text{exo}}$; future exogenous variables $Y^{\text{exo}}$
**Output:** prediction $\hat{Y}^{\text{endo}}$

1: **Training Phase:**
2:   **For** $i = 1$ to $N$ do
3:     **For** $(X_{\text{train}}^{\text{endo}}, X_{\text{train}}^{\text{exo}}, Y_{\text{train}}^{\text{endo}}, Y_{\text{train}}^{\text{exo}})$ in $D$
4:       $z \leftarrow \theta_{\text{Model}}(X_{\text{train}}^{\text{endo}}, X_{\text{train}}^{\text{exo}})$
5:       $\hat{Y}^{\text{endo}} \leftarrow \theta_{\text{MLP}}(\text{Concat}(z, Y_{\text{train}}^{\text{exo}}))$
6:       $\mathcal{L} \leftarrow Loss(\hat{Y}^{\text{endo}}, Y_{\text{train}}^{\text{endo}})$
7:       $\theta_{\text{Model}} \leftarrow \theta_{\text{Model}} - \eta \frac{\partial \mathcal{L}}{\partial \theta_{\text{Model}}}, \ \theta_{\text{MLP}} \leftarrow \theta_{\text{MLP}} - \eta \frac{\partial \mathcal{L}}{\partial \theta_{\text{MLP}}}$
8:     **EndFor**
9:   **EndFor**
10: **Inference Phase:**
11:   $z \leftarrow \theta_{\text{Model}}(X^{\text{endo}}, X^{\text{exo}})$
12:   $\hat{Y}^{\text{endo}} \leftarrow \theta_{\text{MLP}}(\text{Concat}(z, Y^{\text{exo}}))$
13: **return** $\hat{Y}^{\text{endo}}$

---

electric generation is used as the endogenous variable, while the remaining energy types serve as exogenous variables. (2 & 3) **Colbun and Rapel** represent two daily-resolution hydropower datasets from Chile, containing water level, precipitation, and tributary inflow measurements for the Colbún and Rapel reservoirs. The water level is treated as the endogenous variable, while precipitation and inflow are used as exogenous variables. (4-7) **Sdwpf** comprises four wind power generation datasets collected from two turbines in the Longyuan wind farm, including Sdwpfm1 and Sdwpfm2 with hourly resolution, and Sdwpfh1 and Sdwpfh2 with half-hourly resolution. The active power output (Patv) is used as the endogenous variable. The exogenous variables include external weather environment data collected from the ERA5 (Hersbach et al., 2020), such as temperature, surface pressure, relative humidity, wind speed, wind direction, and total precipitation.

## A.2. Implementation Details

1) To keep consistent with previous works, we adopt Mean Squared Error (mse) and Mean Absolute Error (mae) as evaluation metrics.

2) For the 12 real-world datasets that satisfy the TSF-X conditions, we conduct both long-term and short-term prediction experiments. For the Colbun and Raperl datasets, the short-term prediction uses a lookback window of 60 with a prediction horizon of 10, while the long-term prediction uses a lookback window of 180 with a prediction horizon of 30. For the other datasets, the short-term prediction uses a lookback window of 168 with a prediction horizon of 24,

and the long-term prediction uses a lookback window of 720 with a prediction horizon of 360.

3) We utilize the TFB (Qiu et al., 2024) code repository for unified evaluation, with all baseline results also derived from TFB. Following the settings in TFB (Qiu et al., 2024) and TSFM-Bench (Li et al., 2025), we do not apply the "Drop Last" trick to ensure a fair comparison.

4) All experiments of DAG are conducted using Py-Torch (Paszke et al., 2019) in Python 3.8 and executed on an NVIDIA Tesla-A800 GPU. The training process is guided by the L1 loss function and employs the ADAM optimizer. The initial batch size is set to 64, with the flexibility to halve it (down to a minimum of 8) in case of an Out-Of-Memory issue.

### A.3. MLP Fusion Approach

For methods that do not support future covariates, we adapt them to incorporate future exogenous variables using an MLP fusion approach—see Algorithm 1. Note that, for fairness, all methods in our experiments use historical endogenous variables, historical exogenous variables, and future exogenous variables.

## B. Related works

### B.1. Univariate and Multivariate Forecasting

Existing time series forecasting models are typically classified into univariate and multivariate forecasting methods based on the number of input and output variables. **Univariate time series forecasting models** rely solely on the historical values of a single variable to predict future values. Traditional univariate forecasting methods, such as ARIMA (Box & Pierce, 1970), ETS (Hyndman et al., 2008), and Theta (Federico Garza, 2022) are classical and widely used techniques. However, these methods still require manual feature engineering and model design (Fang et al., 2025; Li et al., 2026a; Zhang et al., 2024; 2025b; Ni et al., 2026; Chen et al., 2025; Qi et al., 2025; Ma et al., 2024; 2025b; 2022; Lu et al., 2024b). Leveraging the representation learning of deep neural networks (DNNs) (Fang et al., 2023; 2022; Li et al., 2026b; Zhang et al., 2025a;c; Wang et al., 2026d; Ma et al., 2026; 2025a; Yu et al., 2025d;c;e; Lu et al., 2024a; 2023), many deep learning-based methods emerge. Methods like N-BEATS and DeepAR can automatically learn patterns from historical data, excelling at capturing nonlinear relationships and long-term dependencies. On the other hand, **multivariate time series forecasting models** use multiple input variables to predict the corresponding output variables. The classic methods include VAR (Godahewa et al., 2021), Random Forests(Breiman, 2001) and LightGBM (Ke et al., 2017). In recent years, with the rise of deep learning, various architectural approaches

have gained widespread attention. For instance, Transformer architectures, such as Informer (Zhou et al., 2021), FED-former (Zhou et al., 2022), Autoformer (Wu et al., 2021), Triformer (Cirstea et al., 2022), and PatchTST (Nie et al., 2023), can more accurately capture the complex relationships between temporal tokens. MLP-based methods, such as SparseTSF (Lin et al., 2024a), CycleNet (Lin et al., 2024b), DUET (Qiu et al., 2025c), NLinear (Zeng et al., 2023), and DLinear (Zeng et al., 2023), utilize simpler architectures with fewer parameters but still achieve highly competitive forecasting accuracy. However, all those methods overlook an important practical factor—exogenous (historical or future exogenous). In many real-world scenarios, exogenous data is known or can be approximately known, and utilizing exogenous data can significantly improve the accuracy of predictions.

### B.2. Forecasting with Exogenous Variables

Time series forecasting with exogenous variables has been extensively discussed in classical statistical methods. Some statistical methods have been extended to incorporate exogenous variables as part of the input. Methods like ARI-MAX (Williams, 2001) and SARIMAX (Vagropoulos et al., 2016) have long utilized exogenous variables to enhance forecasting accuracy. More recently, deep learning approaches have advanced this area (Yang et al., 2025a; Hu et al., 2026; Yang et al., 2025b; 2026; Shang et al., 2026; 2024; Chen et al., 2023; Shang & Chen, 2024): CrossLinear (Zhou et al., 2025) uses cross-correlation embeddings to capture dependencies between historical endogenous and exogenous variables; NBEATSx (Olivares et al., 2023) extends N-BEATS with dedicated branches to utilize both past and future exogenous inputs; TiDE (Das et al., 2023) employs an MLP-based architecture to integrate static and future covariates by concatenation with endogenous features at each time step. The Temporal Fusion Transformer (TFT) (Lim et al., 2021) integrates historical and current exogenous variables using attention mechanisms. Furthermore, TimeXer(Wang et al., 2024) introduces patch-wise embeddings to flexibly incorporate exogenous covariates without strict temporal alignment. ExoTST (Tayal et al., 2024) utilizes innovative embedding, cross-attention, and cross-temporal fusion within an attention framework to robustly handle time lags and missing data. On the other hand, GCGNet (Li et al., 2026c) models correlations using a graph structure, but does not explicitly distinguish past and future variables or endogenous and exogenous variables. However, these methods generally rely on relatively simple combinations of historical inputs and future exogenous, without fully modeling the complex interactions among historical endogenous, historical exogenous, and future exogenous variables.

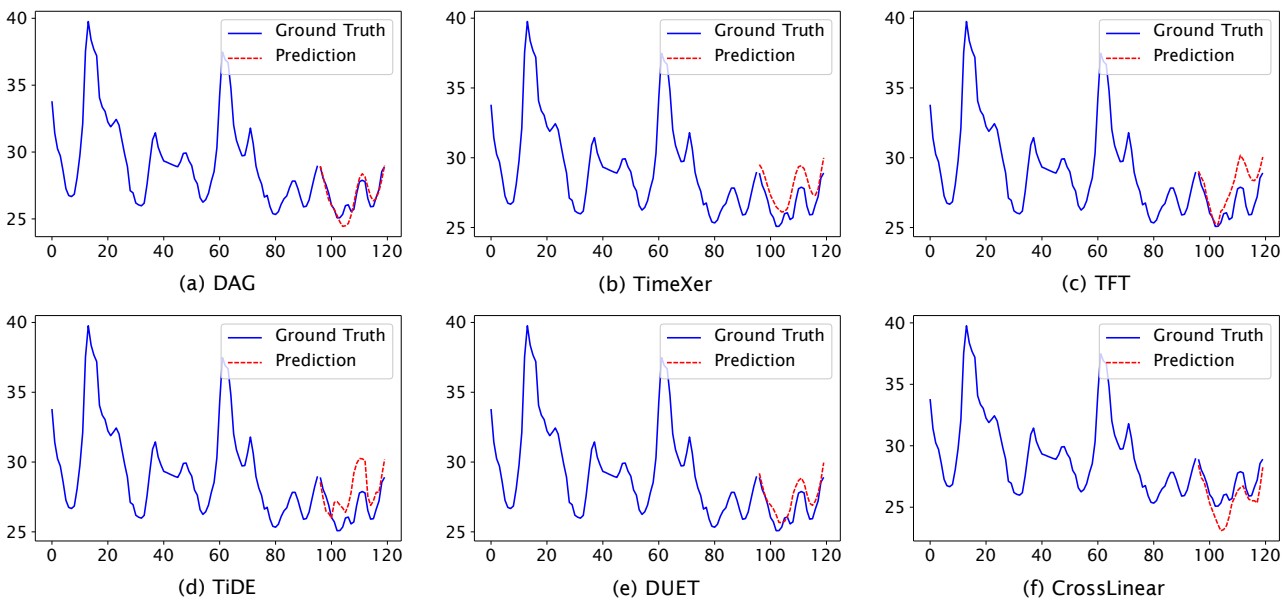

*Figure 6.* Visualization of prediction results on NP dataset where future exogenous variables are available.

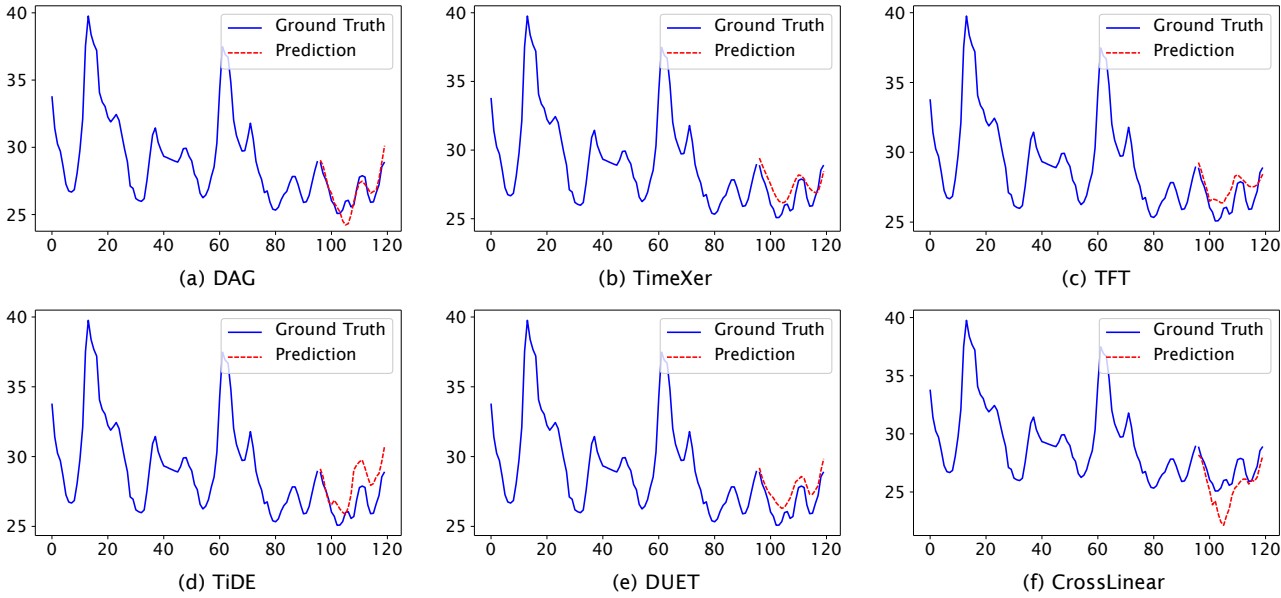

*Figure 7.* Visualization of prediction results on NP dataset where future exogenous variables are not available.

## C. Experimental Results

### C.1. Visualization of Results

We visualize cases of predictions on the NP dataset in Figure 6, where future exogenous variables are available, and Figure 7, where future exogenous variables are not available. As shown in Figure 6 and Figure 7, DAG consistently outperforms all baselines, both when future exogenous variables are available and when they are not, indicating that it effectively discovers and injects correlation relationships across both temporal and channel dimensions, fully leveraging exogenous variables.

### C.2. Full Results

Full results on the 12 multivariate datasets under the TSF-X setting, where the inputs are $(X^{\text{endo}}, X^{\text{exo}}, Y^{\text{exo}})$, are reported in Table 5. Full results on the multivariate datasets without future exogenous variables, where the inputs are $(X^{\text{endo}}, X^{\text{exo}})$, are provided in Table 6.

*Table 5.* Results on 12 real-world datasets that satisfy the TSF-X task, where the inputs are ($X^{\text{endo}}$, $X^{\text{exo}}$, and $Y^{\text{exo}}$). **Red**: the best, Blue: the 2nd best. Avg means the average results from two forecasting horizons.

| Models Metrics | | DAG (ours) mse | mae | GCGNet mse | mae | TimeXer mse | mae | TFT mse | mae | TiDE mse | mae | DUET mse | mae | CrossLinear mse | mae | Amplifier mse | mae | TimeKAN mse | mae | PatchTST mse | mae |
|---|---|---|---|---|---|---|---|---|---|---|---|---|---|---|---|---|---|---|---|---|---|
| NP | 24 | **0.202** | **0.237** | 0.208 | 0.237 | 0.236 | 0.266 | 0.219 | 0.249 | 0.284 | 0.301 | 0.246 | 0.287 | 0.210 | 0.266 | 0.252 | 0.303 | 0.273 | 0.310 | 0.249 | 0.294 |
| | 360 | **0.521** | **0.451** | 0.531 | 0.459 | 0.600 | 0.475 | 0.539 | 0.501 | 0.601 | 0.498 | 0.576 | 0.528 | 0.531 | 0.508 | 0.587 | 0.534 | 0.538 | 0.529 | 0.530 | 0.498 |
| | Avg | **0.362** | **0.344** | 0.370 | 0.348 | 0.418 | 0.371 | 0.379 | 0.375 | 0.443 | 0.400 | 0.411 | 0.408 | 0.371 | 0.387 | 0.420 | 0.418 | 0.405 | 0.419 | 0.390 | 0.396 |
| PJM | 24 | **0.057** | **0.143** | 0.060 | 0.150 | 0.075 | 0.166 | 0.095 | 0.195 | 0.106 | 0.214 | 0.072 | 0.166 | 0.088 | 0.191 | 0.096 | 0.208 | 0.115 | 0.244 | 0.116 | 0.239 |
| | 360 | 0.130 | **0.218** | **0.129** | 0.223 | 0.140 | 0.231 | 0.133 | 0.219 | 0.177 | 0.279 | 0.131 | 0.228 | 0.135 | 0.254 | 0.177 | 0.285 | 0.162 | 0.281 | 0.149 | 0.279 |
| | Avg | **0.093** | **0.180** | 0.095 | 0.187 | 0.108 | 0.198 | 0.114 | 0.207 | 0.142 | 0.246 | 0.102 | 0.197 | 0.112 | 0.223 | 0.137 | 0.246 | 0.139 | 0.262 | 0.133 | 0.259 |
| BE | 24 | 0.361 | **0.229** | **0.350** | 0.248 | 0.392 | 0.253 | 0.426 | 0.272 | 0.426 | 0.285 | 0.432 | 0.272 | 0.391 | 0.259 | 0.471 | 0.339 | 0.451 | 0.319 | 0.452 | 0.326 |
| | 360 | **0.485** | 0.330 | 0.511 | 0.340 | 0.512 | 0.327 | **0.482** | **0.310** | 0.571 | 0.364 | 0.597 | 0.436 | 0.568 | 0.416 | 0.646 | 0.487 | 0.645 | 0.495 | 0.702 | 0.538 |
| | Avg | **0.423** | **0.279** | 0.431 | 0.294 | 0.452 | 0.290 | 0.454 | 0.291 | 0.498 | 0.325 | 0.515 | 0.354 | 0.479 | 0.337 | 0.559 | 0.413 | 0.548 | 0.407 | 0.577 | 0.432 |
| FR | 24 | 0.355 | **0.171** | **0.347** | 0.188 | 0.366 | 0.208 | 0.543 | 0.253 | 0.418 | 0.255 | 0.384 | 0.251 | 0.390 | 0.226 | 0.459 | 0.348 | 0.454 | 0.296 | 0.518 | 0.368 |
| | 360 | 0.473 | 0.268 | 0.482 | 0.279 | 0.489 | 0.273 | **0.465** | **0.261** | 0.551 | 0.308 | 0.607 | 0.403 | 0.575 | 0.370 | 0.648 | 0.468 | 0.641 | 0.452 | 0.658 | 0.452 |
| | Avg | **0.414** | **0.219** | 0.415 | 0.234 | 0.427 | 0.241 | 0.504 | 0.257 | 0.484 | 0.281 | 0.496 | 0.327 | 0.483 | 0.298 | 0.554 | 0.408 | 0.547 | 0.374 | 0.588 | 0.410 |
| DE | 24 | **0.277** | **0.322** | 0.280 | 0.331 | 0.339 | 0.362 | 0.380 | 0.383 | 0.367 | 0.383 | 0.376 | 0.378 | 0.387 | 0.396 | 0.394 | 0.407 | 0.399 | 0.412 | 0.413 | 0.412 |
| | 360 | **0.462** | **0.418** | 0.523 | 0.447 | 0.610 | 0.474 | 0.599 | 0.509 | 0.630 | 0.511 | 0.589 | 0.482 | 0.583 | 0.507 | 0.551 | 0.474 | 0.547 | 0.479 | 0.589 | 0.498 |
| | Avg | **0.370** | **0.370** | 0.401 | 0.389 | 0.475 | 0.418 | 0.489 | 0.446 | 0.499 | 0.447 | 0.482 | 0.430 | 0.485 | 0.452 | 0.473 | 0.441 | 0.473 | 0.445 | 0.501 | 0.455 |
| Energy | 24 | **0.079** | **0.215** | 0.081 | 0.221 | 0.122 | 0.273 | 0.093 | 0.235 | 0.103 | 0.248 | 0.117 | 0.283 | 0.241 | 0.418 | 0.138 | 0.306 | 0.135 | 0.298 | 0.239 | 0.390 |
| | 360 | 0.169 | **0.320** | 0.182 | 0.334 | 0.204 | 0.357 | **0.167** | 0.331 | 0.202 | 0.355 | 0.288 | 0.452 | 0.237 | 0.385 | 0.328 | 0.472 | 0.302 | 0.464 | 0.214 | 0.363 |
| | Avg | **0.124** | **0.267** | 0.131 | 0.277 | 0.163 | 0.315 | 0.130 | 0.283 | 0.153 | 0.302 | 0.203 | 0.367 | 0.239 | 0.402 | 0.233 | 0.389 | 0.218 | 0.381 | 0.226 | 0.377 |
| Sdwpfm1 | 24 | **0.351** | **0.400** | 0.376 | 0.415 | 0.558 | 0.533 | 0.366 | 0.421 | 0.474 | 0.488 | 0.551 | 0.564 | 0.355 | 0.473 | 0.364 | 0.445 | 0.418 | 0.503 | 0.374 | 0.454 |
| | 360 | 0.495 | 0.522 | **0.472** | **0.499** | 0.845 | 0.684 | 0.597 | 0.528 | 0.492 | 0.526 | 0.646 | 0.577 | 0.497 | 0.532 | 0.510 | 0.535 | 0.476 | 0.566 | 0.497 | 0.550 |
| | Avg | **0.423** | 0.461 | 0.424 | **0.457** | 0.701 | 0.609 | 0.482 | 0.474 | 0.483 | 0.507 | 0.599 | 0.570 | 0.426 | 0.502 | 0.437 | 0.490 | 0.447 | 0.534 | 0.435 | 0.502 |
| Sdwpfm2 | 24 | **0.372** | **0.414** | 0.421 | 0.441 | 0.627 | 0.570 | 0.411 | 0.458 | 0.461 | 0.492 | 0.445 | 0.452 | 0.477 | 0.536 | 0.394 | 0.462 | 0.474 | 0.538 | 0.418 | 0.492 |
| | 360 | 0.583 | 0.556 | 0.529 | 0.531 | 0.978 | 0.736 | 0.541 | **0.519** | **0.511** | 0.540 | 0.584 | 0.528 | 0.589 | 0.611 | 0.587 | 0.563 | 0.520 | 0.589 | 0.602 | 0.603 |
| | Avg | 0.477 | **0.485** | **0.475** | 0.486 | 0.803 | 0.653 | 0.476 | 0.488 | 0.486 | 0.516 | 0.514 | 0.490 | 0.533 | 0.573 | 0.491 | 0.512 | 0.497 | 0.564 | 0.510 | 0.547 |
| Sdwpfh1 | 24 | 0.408 | 0.438 | 0.435 | 0.486 | 0.651 | 0.587 | **0.401** | **0.460** | 0.434 | 0.489 | 0.527 | 0.513 | 0.548 | 0.585 | 0.576 | 0.627 | 0.511 | 0.582 | 0.422 | 0.505 |
| | 360 | 0.489 | 0.534 | **0.465** | **0.514** | 0.841 | 0.700 | 0.557 | 0.523 | 0.472 | 0.527 | 0.551 | 0.519 | 0.566 | 0.601 | 0.497 | 0.569 | 0.643 | 0.694 | 0.513 | 0.548 |
| | Avg | **0.448** | **0.486** | 0.450 | 0.500 | 0.746 | 0.643 | 0.479 | 0.491 | 0.453 | 0.508 | 0.539 | 0.516 | 0.557 | 0.593 | 0.537 | 0.598 | 0.577 | 0.638 | 0.468 | 0.527 |
| Sdwpfh2 | 24 | **0.438** | **0.465** | 0.473 | 0.506 | 0.820 | 0.677 | 0.474 | 0.493 | 0.579 | 0.553 | 0.629 | 0.563 | 0.468 | 0.540 | 0.473 | 0.533 | 0.580 | 0.614 | 0.457 | 0.527 |
| | 360 | 0.608 | 0.595 | **0.566** | 0.565 | 0.962 | 0.761 | 0.657 | **0.549** | 0.619 | 0.614 | 0.665 | 0.569 | 0.608 | 0.609 | 0.569 | 0.628 | 0.713 | 0.729 | 0.641 | 0.632 |
| | Avg | 0.523 | 0.530 | **0.520** | 0.536 | 0.891 | 0.719 | 0.566 | **0.521** | 0.599 | 0.583 | 0.647 | 0.566 | 0.538 | 0.574 | 0.521 | 0.581 | 0.647 | 0.672 | 0.549 | 0.580 |
| Colbun | 10 | **0.061** | **0.094** | 0.065 | 0.108 | 0.113 | 0.172 | 0.092 | 0.135 | 0.089 | 0.131 | 0.089 | 0.134 | 0.071 | 0.102 | 0.071 | 0.121 | 0.061 | 0.101 | 0.062 | 0.122 |
| | 30 | **0.135** | **0.215** | 0.149 | 0.243 | 0.176 | 0.299 | 0.383 | 0.460 | 0.240 | 0.322 | 0.307 | 0.397 | 0.182 | 0.288 | 0.275 | 0.370 | 0.195 | 0.249 | 0.417 | 0.496 |
| | Avg | **0.098** | **0.154** | 0.107 | 0.175 | 0.145 | 0.235 | 0.238 | 0.297 | 0.164 | 0.227 | 0.198 | 0.266 | 0.126 | 0.195 | 0.173 | 0.246 | 0.128 | 0.175 | 0.239 | 0.309 |
| Rapel | 10 | **0.151** | **0.203** | 0.211 | 0.230 | 0.301 | 0.308 | 0.201 | 0.253 | 0.228 | 0.271 | 0.174 | 0.219 | 0.163 | 0.209 | 0.181 | 0.227 | 0.174 | 0.231 | 0.163 | 0.218 |
| | 30 | **0.309** | 0.408 | 0.401 | **0.384** | 0.387 | 0.416 | 0.409 | 0.414 | 0.411 | 0.432 | 0.365 | 0.432 | 0.340 | 0.417 | 0.333 | 0.416 | 0.325 | 0.390 | 0.374 | 0.445 |
| | Avg | **0.230** | **0.305** | 0.306 | 0.307 | 0.344 | 0.362 | 0.305 | 0.333 | 0.320 | 0.351 | 0.269 | 0.326 | 0.252 | 0.313 | 0.257 | 0.321 | 0.249 | 0.311 | 0.269 | 0.332 |
| 1st Count | | **23** | **27** | 8 | 4 | 0 | 0 | 4 | 5 | 1 | 0 | 0 | 0 | 0 | 0 | 0 | 0 | 0 | 0 | 0 | 0 |

In addition, full results on the 8 common multivariate datasets with inputs ($X^{\text{endo}}, X^{\text{exo}}, Y^{\text{exo}}$) are presented in Table 8, while the corresponding results using inputs ($X^{\text{endo}}, X^{\text{exo}}$) are reported in Table 7.

### C.3. Limitations and Future Works

Despite the promising performance of DAG, this work still has several limitations. First, the current framework mainly focuses on modeling correlations between exogenous and endogenous variables, rather than explicitly capturing causal relationships. Although the learned correlations are effective for forecasting, incorporating causal inference techniques may further improve the interpretability and reliability of the model. Second, the proposed gating mechanism adopts a simple dot-product interaction between MLP-encoded repre-

sentations for efficiency. While this design achieves a good trade-off between performance and computational cost, it may not fully capture more complex and fine-grained interactions among variables. Exploring more advanced fusion and interaction mechanisms remains an important direction for future work. Third, the current framework involves multiple manually tuned weighting coefficients, including the balancing of the three loss functions and the fusion weights of the two forecasting outputs. Developing adaptive or fully automated weighting strategies is an important direction for future work.

*Table 6.* Results on 12 real-world datasets under the setting where future exogenous variables are not available. The inputs are ($X^{\text{endo}}$ and $X^{\text{exo}}$). The best results are **Red**, and the second-best results are Blue. Avg represents the average results across the two forecasting horizons.

| Models | | DAG | | GCGNet | | TimeXer | | TFT | | TiDE | | DUET | | CrossLinear | | Amplifier | | TimeKAN | | PatchTST | |
| Metrics | | mse | mae | mse | mae | mse | mae | mse | mae | mse | mae | mse | mae | mse | mae | mse | mae | mse | mae | mse | mae |
|---|---|---|---|---|---|---|---|---|---|---|---|---|---|---|---|---|---|---|---|---|---|
| NP | 24 | 0.237 | 0.259 | 0.240 | 0.260 | 0.270 | 0.292 | 0.330 | 0.321 | 0.297 | 0.309 | 0.262 | 0.271 | 0.246 | 0.282 | 0.302 | 0.321 | 0.336 | 0.344 | 0.283 | 0.296 |
| | 360 | 0.601 | 0.500 | 0.620 | 0.502 | 0.609 | 0.474 | 0.964 | 0.655 | 0.638 | 0.523 | 0.626 | 0.494 | 0.655 | 0.505 | 0.738 | 0.574 | 0.632 | 0.527 | 0.630 | 0.505 |
| | Avg | 0.419 | 0.380 | 0.430 | 0.381 | 0.440 | 0.383 | 0.647 | 0.488 | 0.467 | 0.416 | 0.444 | 0.383 | 0.451 | 0.394 | 0.520 | 0.448 | 0.484 | 0.435 | 0.457 | 0.401 |
| PJM | 24 | 0.072 | 0.162 | 0.072 | 0.164 | 0.096 | 0.191 | 0.129 | 0.231 | 0.105 | 0.213 | 0.082 | 0.178 | 0.096 | 0.198 | 0.093 | 0.195 | 0.130 | 0.236 | 0.106 | 0.212 |
| | 360 | 0.180 | 0.259 | 0.191 | 0.269 | 0.186 | 0.267 | 0.270 | 0.310 | 0.210 | 0.292 | 0.198 | 0.273 | 0.197 | 0.283 | 0.210 | 0.294 | 0.219 | 0.304 | 0.189 | 0.273 |
| | Avg | 0.126 | 0.210 | 0.131 | 0.216 | 0.141 | 0.229 | 0.200 | 0.270 | 0.158 | 0.253 | 0.140 | 0.226 | 0.147 | 0.241 | 0.152 | 0.245 | 0.175 | 0.270 | 0.148 | 0.243 |
| BE | 24 | 0.376 | 0.244 | 0.365 | 0.259 | 0.392 | 0.253 | 0.400 | 0.267 | 0.492 | 0.322 | 0.376 | 0.253 | 0.389 | 0.253 | 0.417 | 0.283 | 0.424 | 0.280 | 0.405 | 0.264 |
| | 360 | 0.547 | 0.350 | 0.553 | 0.354 | 0.563 | 0.349 | 0.727 | 0.442 | 0.602 | 0.375 | 0.571 | 0.357 | 0.564 | 0.346 | 0.588 | 0.367 | 0.552 | 0.350 | 0.564 | 0.354 |
| | Avg | 0.462 | 0.297 | 0.459 | 0.306 | 0.477 | 0.301 | 0.563 | 0.354 | 0.547 | 0.348 | 0.473 | 0.305 | 0.477 | 0.300 | 0.502 | 0.325 | 0.488 | 0.315 | 0.485 | 0.309 |
| FR | 24 | 0.345 | 0.190 | 0.352 | 0.199 | 0.366 | 0.195 | 0.445 | 0.231 | 0.415 | 0.254 | 0.359 | 0.208 | 0.397 | 0.204 | 0.429 | 0.250 | 0.448 | 0.250 | 0.397 | 0.223 |
| | 360 | 0.526 | 0.307 | 0.519 | 0.293 | 0.542 | 0.300 | 0.624 | 0.345 | 0.574 | 0.325 | 0.577 | 0.317 | 0.555 | 0.310 | 0.559 | 0.322 | 0.534 | 0.303 | 0.542 | 0.306 |
| | Avg | 0.435 | 0.249 | 0.435 | 0.246 | 0.454 | 0.247 | 0.535 | 0.288 | 0.494 | 0.290 | 0.468 | 0.262 | 0.476 | 0.257 | 0.494 | 0.286 | 0.491 | 0.276 | 0.470 | 0.264 |
| DE | 24 | 0.422 | 0.411 | 0.413 | 0.402 | 0.501 | 0.445 | 0.576 | 0.452 | 0.465 | 0.433 | 0.424 | 0.403 | 0.438 | 0.423 | 0.493 | 0.455 | 0.504 | 0.462 | 0.503 | 0.450 |
| | 360 | 0.784 | 0.563 | 0.816 | 0.600 | 0.818 | 0.568 | 0.792 | 0.578 | 0.823 | 0.604 | 0.895 | 0.623 | 0.832 | 0.592 | 0.930 | 0.642 | 0.834 | 0.590 | 0.889 | 0.604 |
| | Avg | 0.603 | 0.487 | 0.614 | 0.501 | 0.659 | 0.507 | 0.684 | 0.515 | 0.644 | 0.519 | 0.660 | 0.513 | 0.635 | 0.508 | 0.712 | 0.548 | 0.669 | 0.526 | 0.696 | 0.527 |
| Energy | 24 | 0.112 | 0.257 | 0.112 | 0.259 | 0.138 | 0.293 | 0.352 | 0.463 | 0.117 | 0.265 | 0.108 | 0.254 | 0.102 | 0.246 | 0.108 | 0.254 | 0.109 | 0.254 | 0.108 | 0.254 |
| | 360 | 0.189 | 0.342 | 0.195 | 0.351 | 0.206 | 0.360 | 0.399 | 0.497 | 0.207 | 0.358 | 0.211 | 0.362 | 0.299 | 0.425 | 0.251 | 0.400 | 0.186 | 0.338 | 0.298 | 0.428 |
| | Avg | 0.150 | 0.300 | 0.154 | 0.305 | 0.172 | 0.326 | 0.376 | 0.480 | 0.162 | 0.311 | 0.160 | 0.308 | 0.201 | 0.336 | 0.180 | 0.327 | 0.147 | 0.296 | 0.203 | 0.341 |
| Sdwpfm1 | 24 | 0.532 | 0.512 | 0.537 | 0.509 | 0.558 | 0.533 | 0.596 | 0.531 | 0.561 | 0.540 | 0.550 | 0.495 | 0.549 | 0.553 | 0.579 | 0.557 | 0.598 | 0.579 | 0.557 | 0.557 |
| | 360 | 0.845 | 0.695 | 0.885 | 0.682 | 0.845 | 0.684 | 1.372 | 0.925 | 0.864 | 0.725 | 0.898 | 0.672 | 1.070 | 0.834 | 1.106 | 0.814 | 0.853 | 0.764 | 0.910 | 0.745 |
| | Avg | 0.689 | 0.604 | 0.711 | 0.595 | 0.701 | 0.609 | 0.984 | 0.728 | 0.713 | 0.633 | 0.724 | 0.583 | 0.809 | 0.694 | 0.843 | 0.685 | 0.725 | 0.672 | 0.733 | 0.651 |
| Sdwpfm2 | 24 | 0.609 | 0.548 | 0.630 | 0.547 | 0.627 | 0.570 | 0.749 | 0.657 | 0.636 | 0.582 | 0.612 | 0.537 | 0.635 | 0.586 | 0.669 | 0.600 | 0.684 | 0.621 | 0.648 | 0.610 |
| | 360 | 0.962 | 0.766 | 0.983 | 0.739 | 0.978 | 0.736 | 1.340 | 0.877 | 0.995 | 0.783 | 1.028 | 0.745 | 0.965 | 0.788 | 1.215 | 0.863 | 0.987 | 0.780 | 1.020 | 0.819 |
| | Avg | 0.786 | 0.657 | 0.807 | 0.643 | 0.803 | 0.653 | 1.044 | 0.767 | 0.816 | 0.682 | 0.820 | 0.641 | 0.800 | 0.687 | 0.942 | 0.732 | 0.836 | 0.701 | 0.834 | 0.714 |
| Sdwpfh1 | 24 | 0.647 | 0.592 | 0.651 | 0.578 | 0.651 | 0.587 | 0.754 | 0.670 | 0.719 | 0.641 | 0.673 | 0.575 | 0.687 | 0.652 | 1.111 | 0.816 | 0.755 | 0.679 | 0.669 | 0.629 |
| | 360 | 0.820 | 0.695 | 0.830 | 0.677 | 0.841 | 0.700 | 0.831 | 0.727 | 0.897 | 0.757 | 0.886 | 0.712 | 0.849 | 0.752 | 1.089 | 0.823 | 0.853 | 0.762 | 0.939 | 0.804 |
| | Avg | 0.733 | 0.643 | 0.741 | 0.628 | 0.746 | 0.643 | 0.793 | 0.698 | 0.808 | 0.699 | 0.779 | 0.644 | 0.768 | 0.702 | 1.100 | 0.820 | 0.804 | 0.720 | 0.804 | 0.717 |
| Sdwpfh2 | 24 | 0.739 | 0.635 | 0.798 | 0.645 | 0.820 | 0.677 | 0.811 | 0.714 | 0.783 | 0.674 | 0.975 | 0.694 | 0.810 | 0.718 | 0.834 | 0.691 | 0.892 | 0.740 | 0.769 | 0.680 |
| | 360 | 1.000 | 0.755 | 0.974 | 0.757 | 0.962 | 0.761 | 1.042 | 0.779 | 1.056 | 0.829 | 1.039 | 0.735 | 1.102 | 0.830 | 1.126 | 0.845 | 0.990 | 0.825 | 1.280 | 0.925 |
| | Avg | 0.870 | 0.695 | 0.886 | 0.701 | 0.891 | 0.719 | 0.926 | 0.746 | 0.919 | 0.751 | 1.007 | 0.715 | 0.956 | 0.774 | 0.980 | 0.768 | 0.941 | 0.782 | 1.025 | 0.802 |
| Colbun | 10 | 0.072 | 0.103 | 0.070 | 0.094 | 0.084 | 0.146 | 0.449 | 0.279 | 0.095 | 0.140 | 0.073 | 0.106 | 0.068 | 0.101 | 0.077 | 0.120 | 0.091 | 0.129 | 0.083 | 0.125 |
| | 30 | 0.161 | 0.208 | 0.168 | 0.236 | 0.181 | 0.293 | 0.663 | 0.494 | 0.282 | 0.341 | 0.221 | 0.290 | 0.190 | 0.289 | 0.227 | 0.301 | 0.311 | 0.377 | 0.217 | 0.316 |
| | Avg | 0.117 | 0.155 | 0.119 | 0.165 | 0.132 | 0.219 | 0.556 | 0.386 | 0.188 | 0.240 | 0.147 | 0.198 | 0.129 | 0.195 | 0.152 | 0.210 | 0.201 | 0.253 | 0.150 | 0.221 |
| Rapel | 10 | 0.195 | 0.215 | 0.217 | 0.221 | 0.301 | 0.308 | 0.238 | 0.274 | 0.238 | 0.270 | 0.221 | 0.225 | 0.201 | 0.224 | 0.240 | 0.259 | 0.226 | 0.238 | 0.215 | 0.232 |
| | 30 | 0.327 | 0.347 | 0.313 | 0.346 | 0.387 | 0.416 | 0.377 | 0.394 | 0.409 | 0.444 | 0.387 | 0.386 | 0.278 | 0.374 | 0.433 | 0.436 | 0.458 | 0.437 | 0.436 | 0.436 |
| | Avg | 0.261 | 0.281 | 0.265 | 0.284 | 0.344 | 0.362 | 0.308 | 0.334 | 0.323 | 0.357 | 0.304 | 0.306 | 0.240 | 0.299 | 0.337 | 0.348 | 0.342 | 0.337 | 0.325 | 0.334 |
| 1st Count | | 25 | 16 | 4 | 7 | 1 | 2 | 0 | 0 | 0 | 0 | 0 | 7 | 4 | 2 | 0 | 0 | 2 | 2 | 0 | 0 |

*Table 7.* Results on 8 common datasets under the setting where future exogenous variables are not available. The inputs are ($X^{endo}$ and $X^{exo}$). The best results are **Red**, and the second-best results are Blue. Avg represents the average results across forecasting horizons.

| Models | | DAG | | TimeXer | | TFT | | TiDE | | DUET | | CrossLinear | | Amplifier | | TimeKAN | | PatchTST | |
|---|---|---|---|---|---|---|---|---|---|---|---|---|---|---|---|---|---|---|---|
| Metrics | | mse | mae | mse | mae | mse | mae | mse | mae | mse | mae | mse | mae | mse | mae | mse | mae | mse | mae |
| ETTh1 | 96 | 0.061 | 0.188 | 0.057 | 0.181 | 0.068 | 0.200 | 0.058 | 0.182 | 0.058 | 0.183 | 0.056 | 0.181 | 0.057 | 0.181 | **0.055** | **0.178** | 0.056 | 0.179 |
| | 192 | **0.070** | 0.206 | 0.074 | 0.208 | 0.117 | 0.267 | 0.075 | 0.210 | 0.074 | 0.207 | 0.073 | 0.209 | 0.072 | **0.205** | 0.074 | 0.209 | 0.075 | 0.210 |
| | 336 | **0.082** | 0.227 | 0.087 | 0.231 | 0.152 | 0.304 | 0.090 | 0.234 | 0.092 | 0.237 | 0.083 | **0.226** | 0.089 | 0.233 | 0.090 | 0.236 | 0.087 | 0.231 |
| | 720 | **0.079** | **0.225** | 0.096 | 0.243 | 0.143 | 0.302 | 0.096 | 0.242 | 0.124 | 0.276 | 0.082 | 0.226 | 0.090 | 0.236 | 0.093 | 0.241 | 0.091 | 0.238 |
| | Avg | **0.073** | 0.211 | 0.078 | 0.216 | 0.120 | 0.268 | 0.079 | 0.217 | 0.087 | 0.226 | 0.074 | **0.210** | 0.077 | 0.214 | 0.078 | 0.216 | 0.077 | 0.214 |
| ETTh2 | 96 | 0.134 | 0.288 | 0.137 | 0.285 | 0.161 | 0.315 | **0.127** | **0.271** | 0.130 | 0.277 | 0.137 | 0.289 | 0.133 | 0.280 | 0.128 | 0.274 | 0.145 | 0.295 |
| | 192 | **0.163** | **0.320** | 0.189 | 0.343 | 0.208 | 0.360 | 0.178 | 0.327 | 0.182 | 0.333 | 0.185 | 0.338 | 0.181 | 0.333 | 0.179 | 0.330 | 0.189 | 0.342 |
| | 336 | **0.183** | **0.350** | 0.225 | 0.380 | 0.258 | 0.407 | 0.223 | 0.376 | 0.219 | 0.374 | 0.220 | 0.376 | 0.217 | 0.371 | 0.222 | 0.376 | 0.231 | 0.386 |
| | 720 | **0.172** | **0.338** | 0.250 | 0.399 | 0.314 | 0.456 | 0.260 | 0.411 | 0.244 | 0.397 | 0.255 | 0.405 | 0.238 | 0.392 | 0.250 | 0.402 | 0.234 | 0.388 |
| | Avg | **0.163** | **0.324** | 0.200 | 0.352 | 0.235 | 0.385 | 0.197 | 0.346 | 0.194 | 0.345 | 0.200 | 0.352 | 0.192 | 0.344 | 0.195 | 0.345 | 0.200 | 0.353 |
| ETTm1 | 96 | **0.029** | 0.127 | 0.029 | **0.126** | 0.037 | 0.146 | 0.043 | 0.159 | 0.030 | 0.128 | 0.029 | 0.127 | 0.031 | 0.131 | 0.030 | 0.130 | 0.029 | 0.127 |
| | 192 | **0.044** | **0.159** | 0.044 | 0.159 | 0.060 | 0.188 | 0.050 | 0.172 | 0.047 | 0.162 | 0.044 | 0.160 | 0.045 | 0.162 | 0.044 | 0.160 | 0.045 | 0.160 |
| | 336 | **0.057** | **0.184** | 0.058 | 0.186 | 0.072 | 0.208 | 0.068 | 0.200 | 0.071 | 0.198 | 0.058 | 0.185 | 0.057 | 0.185 | 0.057 | 0.185 | 0.058 | 0.185 |
| | 720 | **0.079** | **0.216** | 0.081 | 0.219 | 0.119 | 0.272 | 0.079 | 0.218 | 0.088 | 0.225 | 0.082 | 0.218 | 0.080 | 0.219 | 0.080 | 0.218 | 0.082 | 0.220 |
| | Avg | **0.052** | **0.171** | 0.053 | 0.172 | 0.072 | 0.204 | 0.060 | 0.187 | 0.059 | 0.178 | 0.053 | 0.173 | 0.053 | 0.174 | 0.053 | 0.173 | 0.053 | 0.173 |
| ETTm2 | 96 | **0.062** | **0.179** | 0.068 | 0.192 | 0.264 | 0.389 | 0.070 | 0.190 | 0.066 | 0.182 | 0.064 | 0.180 | 0.067 | 0.187 | 0.075 | 0.201 | 0.065 | 0.183 |
| | 192 | **0.095** | **0.229** | 0.104 | 0.243 | 0.338 | 0.443 | 0.100 | 0.235 | 0.098 | 0.230 | 0.098 | 0.232 | 0.102 | 0.237 | 0.105 | 0.244 | 0.101 | 0.236 |
| | 336 | **0.125** | **0.267** | 0.130 | 0.276 | 0.272 | 0.407 | 0.128 | 0.272 | 0.126 | 0.268 | 0.130 | 0.273 | 0.134 | 0.279 | 0.132 | 0.278 | 0.130 | 0.273 |
| | 720 | **0.174** | **0.322** | 0.185 | 0.335 | 0.339 | 0.466 | 0.179 | 0.328 | 0.178 | 0.326 | 0.182 | 0.331 | 0.183 | 0.334 | 0.181 | 0.331 | 0.183 | 0.331 |
| | Avg | **0.114** | **0.249** | 0.122 | 0.261 | 0.303 | 0.427 | 0.119 | 0.256 | 0.117 | 0.251 | 0.119 | 0.254 | 0.122 | 0.259 | 0.123 | 0.263 | 0.120 | 0.256 |
| Weather | 96 | **0.001** | **0.023** | 0.001 | 0.024 | 0.001 | 0.028 | 0.001 | 0.026 | 0.001 | 0.024 | 0.001 | 0.028 | 0.001 | 0.028 | 0.001 | 0.028 | 0.001 | 0.028 |
| | 192 | **0.001** | **0.026** | 0.001 | 0.026 | 0.002 | 0.029 | 0.002 | 0.029 | 0.001 | 0.026 | 0.002 | 0.031 | 0.002 | 0.030 | 0.002 | 0.031 | 0.002 | 0.030 |
| | 336 | **0.001** | **0.026** | 0.001 | 0.028 | 0.002 | 0.030 | 0.002 | 0.030 | 0.001 | 0.028 | 0.002 | 0.032 | 0.002 | 0.032 | 0.002 | 0.032 | 0.002 | 0.032 |
| | 720 | **0.002** | **0.030** | 0.002 | 0.032 | 0.002 | 0.034 | 0.002 | 0.035 | 0.002 | 0.032 | 0.002 | 0.036 | 0.002 | 0.036 | 0.002 | 0.036 | 0.002 | 0.036 |
| | Avg | **0.001** | **0.026** | 0.001 | 0.027 | 0.002 | 0.030 | 0.002 | 0.030 | 0.001 | 0.028 | 0.002 | 0.031 | 0.002 | 0.032 | 0.002 | 0.032 | 0.002 | 0.032 |
| Electricity | 96 | 0.318 | 0.417 | 0.305 | 0.396 | 0.361 | 0.448 | 0.286 | **0.377** | 0.292 | 0.380 | **0.271** | 0.377 | 0.376 | 0.445 | 0.459 | 0.498 | 0.300 | 0.384 |
| | 192 | **0.299** | 0.395 | 0.333 | 0.412 | 0.429 | 0.488 | 0.321 | 0.396 | 0.315 | **0.391** | 0.308 | 0.392 | 0.388 | 0.448 | 0.407 | 0.459 | 0.339 | 0.408 |
| | 336 | **0.336** | 0.426 | 0.388 | 0.446 | 0.438 | 0.492 | 0.340 | **0.421** | 0.367 | 0.425 | 0.365 | 0.435 | 0.438 | 0.477 | 0.464 | 0.495 | 0.385 | 0.434 |
| | 720 | **0.396** | **0.469** | 0.456 | 0.493 | 0.464 | 0.506 | 0.508 | 0.486 | 0.459 | 0.491 | 0.413 | 0.473 | 0.503 | 0.525 | 0.568 | 0.562 | 0.466 | 0.494 |
| | Avg | **0.337** | 0.427 | 0.371 | 0.437 | 0.423 | 0.484 | 0.364 | 0.420 | 0.358 | 0.422 | 0.339 | **0.419** | 0.426 | 0.474 | 0.474 | 0.503 | 0.372 | 0.430 |
| Traffic | 96 | **0.164** | **0.240** | 0.183 | 0.279 | 0.168 | 0.269 | 0.250 | 0.364 | 0.230 | 0.314 | 0.174 | 0.261 | 0.173 | 0.259 | 0.220 | 0.311 | 0.204 | 0.291 |
| | 192 | **0.159** | **0.230** | 0.180 | 0.278 | 0.178 | 0.272 | 0.206 | 0.327 | 0.210 | 0.297 | 0.167 | 0.250 | 0.166 | 0.248 | 0.193 | 0.280 | 0.190 | 0.279 |
| | 336 | **0.157** | **0.235** | 0.180 | 0.284 | 0.169 | 0.263 | 0.261 | 0.395 | 0.205 | 0.297 | 0.164 | 0.251 | 0.162 | 0.248 | 0.198 | 0.285 | 0.183 | 0.276 |
| | 720 | 0.188 | 0.266 | 0.199 | 0.300 | **0.166** | **0.264** | 0.316 | 0.459 | 0.229 | 0.321 | 0.179 | 0.271 | 0.185 | 0.269 | 0.219 | 0.307 | 0.203 | 0.292 |
| | Avg | **0.167** | **0.243** | 0.186 | 0.285 | 0.170 | 0.267 | 0.259 | 0.386 | 0.218 | 0.307 | 0.171 | 0.258 | 0.171 | 0.256 | 0.207 | 0.296 | 0.195 | 0.284 |
| Exchange | 96 | 0.109 | 0.260 | 0.107 | 0.242 | 0.112 | 0.251 | 0.102 | 0.237 | 0.108 | 0.240 | 0.103 | 0.232 | 0.116 | 0.260 | 0.125 | 0.278 | **0.097** | **0.228** |
| | 192 | **0.199** | 0.360 | 0.226 | 0.354 | 0.228 | 0.366 | 0.208 | 0.343 | 0.210 | 0.346 | 0.212 | **0.340** | 0.225 | 0.369 | 0.232 | 0.385 | 0.221 | 0.345 |
| | 336 | **0.285** | **0.427** | 0.417 | 0.483 | 0.461 | 0.520 | 0.459 | 0.503 | 0.399 | 0.488 | 0.398 | 0.482 | 0.430 | 0.497 | 0.438 | 0.504 | 0.424 | 0.486 |
| | 720 | **0.545** | **0.605** | 1.224 | 0.847 | 1.201 | 0.841 | 1.169 | 0.832 | 0.869 | 0.726 | 1.078 | 0.791 | 1.083 | 0.802 | 1.065 | 0.800 | 1.119 | 0.807 |
| | Avg | **0.284** | **0.413** | 0.494 | 0.481 | 0.500 | 0.494 | 0.484 | 0.479 | 0.396 | 0.450 | 0.448 | 0.462 | 0.463 | 0.482 | 0.465 | 0.492 | 0.465 | 0.467 |
| 1st Count | | **35** | **27** | 0 | 1 | 1 | 1 | 1 | 3 | 0 | 1 | 1 | 4 | 0 | 1 | 1 | 1 | 1 | 1 |

*Table 8.* Results on 8 common datasets that satisfy the TSF-X task, where the inputs are ($X^{\text{endo}}$, $X^{\text{exo}}$, and $Y^{\text{exo}}$). **Red**: the best, Blue: the 2nd best. Avg means the average results from forecasting horizons.

| Models | | DAG (ours) | | TimeXer | | TFT | | TiDE | | DUET | | CrossLinear | | Amplifier | | TimeKAN | | xPatch | | PatchTST | |
|---|---|---|---|---|---|---|---|---|---|---|---|---|---|---|---|---|---|---|---|---|---|
| Metrics | | mse | mae | mse | mae | mse | mae | mse | mae | mse | mae | mse | mae | mse | mae | mse | mae | mse | mae | mse | mae |
| ETTh1 | 96 | **0.054** | **0.176** | 0.056 | 0.179 | 0.081 | 0.214 | 0.056 | 0.179 | 0.163 | 0.338 | 0.152 | 0.321 | 0.160 | 0.332 | 0.118 | 0.273 | 0.136 | 0.305 | 0.216 | 0.403 |
| | 192 | **0.072** | 0.208 | 0.073 | **0.207** | 0.073 | 0.213 | 0.075 | 0.210 | 0.206 | 0.382 | 0.199 | 0.374 | 0.134 | 0.290 | 0.166 | 0.330 | 0.150 | 0.311 | 0.174 | 0.336 |
| | 336 | **0.079** | **0.219** | 0.089 | 0.234 | 0.099 | 0.246 | 0.090 | 0.236 | 0.248 | 0.419 | 0.218 | 0.383 | 0.162 | 0.325 | 0.140 | 0.295 | 0.133 | 0.291 | 0.253 | 0.429 |
| | 720 | **0.079** | **0.221** | 0.104 | 0.254 | 0.124 | 0.282 | 0.096 | 0.244 | 0.283 | 0.446 | 0.295 | 0.475 | 0.263 | 0.429 | 0.284 | 0.440 | 0.239 | 0.406 | 0.302 | 0.462 |
| | Avg | **0.071** | **0.206** | 0.081 | 0.219 | 0.094 | 0.239 | 0.079 | 0.217 | 0.225 | 0.396 | 0.216 | 0.388 | 0.180 | 0.344 | 0.177 | 0.335 | 0.165 | 0.328 | 0.236 | 0.407 |
| ETTh2 | 96 | 0.127 | 0.273 | 0.141 | 0.289 | 0.217 | 0.364 | 0.127 | **0.271** | 0.121 | 0.275 | 0.135 | 0.289 | **0.119** | 0.275 | 0.122 | 0.278 | 0.191 | 0.352 | 0.134 | 0.293 |
| | 192 | 0.177 | 0.327 | 0.194 | 0.346 | 0.256 | 0.405 | 0.179 | 0.327 | 0.234 | 0.391 | 0.154 | 0.323 | 0.200 | 0.356 | 0.186 | 0.354 | **0.142** | **0.305** | 0.158 | 0.326 |
| | 336 | 0.215 | 0.369 | 0.226 | 0.380 | 0.287 | 0.435 | 0.223 | 0.375 | 0.228 | 0.379 | 0.179 | 0.343 | 0.209 | 0.367 | 0.181 | 0.342 | 0.185 | 0.349 | **0.172** | **0.338** |
| | 720 | 0.214 | 0.370 | 0.234 | 0.385 | 0.283 | 0.431 | 0.253 | 0.405 | **0.209** | **0.365** | 0.319 | 0.463 | 0.348 | 0.488 | 0.337 | 0.473 | 0.223 | 0.385 | 0.337 | 0.475 |
| | Avg | **0.183** | **0.335** | 0.199 | 0.350 | 0.261 | 0.409 | 0.195 | 0.344 | 0.198 | 0.352 | 0.197 | 0.354 | 0.219 | 0.372 | 0.207 | 0.362 | 0.185 | 0.348 | 0.200 | 0.358 |
| ETTm1 | 96 | **0.029** | **0.127** | 0.030 | 0.129 | 0.030 | 0.131 | 0.030 | 0.130 | 0.063 | 0.204 | 0.087 | 0.247 | 0.092 | 0.252 | 0.067 | 0.205 | 0.055 | 0.188 | 0.092 | 0.253 |
| | 192 | **0.044** | **0.159** | 0.044 | 0.161 | 0.048 | 0.168 | 0.049 | 0.168 | 0.217 | 0.413 | 0.164 | 0.351 | 0.177 | 0.359 | 0.156 | 0.331 | 0.128 | 0.300 | 0.132 | 0.304 |
| | 336 | **0.057** | **0.184** | 0.060 | 0.189 | 0.068 | 0.200 | 0.069 | 0.203 | 0.252 | 0.446 | 0.239 | 0.426 | 0.247 | 0.435 | 0.191 | 0.363 | 0.155 | 0.333 | 0.255 | 0.440 |
| | 720 | 0.079 | 0.216 | 0.082 | 0.222 | 0.095 | 0.240 | **0.073** | **0.211** | 0.265 | 0.436 | 0.286 | 0.462 | 0.252 | 0.424 | 0.261 | 0.428 | 0.247 | 0.425 | 0.251 | 0.434 |
| | Avg | **0.052** | **0.172** | 0.054 | 0.175 | 0.060 | 0.185 | 0.055 | 0.178 | 0.199 | 0.375 | 0.194 | 0.372 | 0.192 | 0.367 | 0.169 | 0.332 | 0.146 | 0.312 | 0.183 | 0.358 |
| ETTm2 | 96 | **0.063** | **0.183** | 0.072 | 0.195 | 0.117 | 0.253 | 0.070 | 0.190 | 0.107 | 0.260 | 0.156 | 0.332 | 0.175 | 0.353 | 0.146 | 0.309 | 0.151 | 0.320 | 0.173 | 0.351 |
| | 192 | 0.098 | **0.230** | 0.110 | 0.249 | 0.179 | 0.324 | 0.101 | 0.235 | 0.200 | 0.366 | 0.203 | 0.366 | 0.179 | 0.339 | 0.139 | 0.293 | **0.096** | 0.235 | 0.142 | 0.295 |
| | 336 | **0.127** | **0.270** | 0.139 | 0.284 | 0.206 | 0.356 | 0.128 | 0.272 | 0.136 | 0.292 | 0.136 | 0.292 | 0.139 | 0.294 | 0.182 | 0.338 | 0.141 | 0.291 | 0.151 | 0.311 |
| | 720 | 0.179 | 0.328 | 0.190 | 0.340 | 0.271 | 0.417 | 0.179 | 0.328 | **0.147** | **0.310** | 0.156 | 0.316 | 0.175 | 0.332 | 0.169 | 0.332 | 0.204 | 0.362 | 0.164 | 0.326 |
| | Avg | **0.117** | **0.253** | 0.128 | 0.267 | 0.193 | 0.338 | 0.119 | 0.256 | 0.148 | 0.307 | 0.163 | 0.326 | 0.167 | 0.330 | 0.159 | 0.318 | 0.148 | 0.302 | 0.157 | 0.321 |
| Weather | 96 | **0.001** | **0.023** | 0.001 | 0.023 | 0.001 | 0.028 | 0.001 | 0.026 | 0.002 | 0.028 | 0.012 | 0.093 | 0.008 | 0.074 | 0.008 | 0.072 | 0.001 | 0.027 | 0.006 | 0.062 |
| | 192 | **0.001** | **0.022** | 0.001 | 0.026 | 0.002 | 0.030 | 0.001 | 0.028 | 0.002 | 0.030 | 0.005 | 0.056 | 0.008 | 0.073 | 0.009 | 0.078 | 0.001 | 0.024 | 0.008 | 0.073 |
| | 336 | 0.002 | 0.028 | 0.001 | 0.027 | 0.002 | 0.030 | 0.002 | 0.031 | 0.001 | **0.025** | 0.009 | 0.080 | 0.008 | 0.071 | 0.009 | 0.074 | 0.002 | 0.029 | 0.011 | 0.087 |
| | 720 | **0.002** | **0.030** | 0.002 | 0.032 | 0.002 | 0.035 | 0.002 | 0.036 | 0.002 | 0.033 | 0.007 | 0.067 | 0.019 | 0.113 | 0.003 | 0.045 | 0.002 | 0.037 | 0.012 | 0.084 |
| | Avg | 0.002 | **0.026** | 0.001 | 0.027 | 0.002 | 0.031 | 0.002 | 0.030 | 0.001 | 0.029 | 0.008 | 0.074 | 0.011 | 0.083 | 0.007 | 0.067 | 0.002 | 0.029 | 0.009 | 0.076 |
| Electricity | 96 | **0.158** | 0.298 | 0.160 | **0.296** | 0.365 | 0.456 | 0.312 | 0.414 | 0.248 | 0.385 | 0.281 | 0.416 | 0.279 | 0.416 | 0.276 | 0.413 | 0.249 | 0.387 | 0.274 | 0.406 |
| | 192 | **0.156** | **0.295** | 0.186 | 0.320 | 0.339 | 0.448 | 0.274 | 0.391 | 0.290 | 0.421 | 0.248 | 0.384 | 0.296 | 0.427 | 0.309 | 0.435 | 0.258 | 0.397 | 0.264 | 0.399 |
| | 336 | **0.217** | 0.353 | 0.218 | **0.346** | 0.367 | 0.471 | 0.313 | 0.417 | 0.295 | 0.426 | 0.260 | 0.396 | 0.278 | 0.414 | 0.292 | 0.419 | 0.296 | 0.425 | 0.271 | 0.408 |
| | 720 | **0.246** | **0.370** | 0.267 | 0.386 | 0.429 | 0.510 | 0.353 | 0.445 | 0.290 | 0.414 | 0.268 | 0.401 | 0.255 | 0.396 | 0.327 | 0.448 | 0.256 | 0.392 | 0.268 | 0.402 |
| | Avg | **0.194** | **0.329** | 0.208 | 0.337 | 0.375 | 0.471 | 0.313 | 0.417 | 0.281 | 0.411 | 0.264 | 0.399 | 0.277 | 0.413 | 0.301 | 0.429 | 0.265 | 0.400 | 0.269 | 0.404 |
| Traffic | 96 | **0.119** | **0.187** | 0.135 | 0.233 | 0.139 | 0.221 | 0.149 | 0.251 | 0.247 | 0.316 | 0.261 | 0.322 | 0.174 | 0.240 | 0.221 | 0.292 | 0.244 | 0.305 | 0.241 | 0.307 |
| | 192 | 0.121 | **0.191** | 0.135 | 0.230 | **0.116** | 0.212 | 0.149 | 0.246 | 0.240 | 0.297 | 0.232 | 0.289 | 0.214 | 0.276 | 0.181 | 0.254 | 0.220 | 0.292 | 0.246 | 0.305 |
| | 336 | **0.125** | **0.200** | 0.143 | 0.237 | 0.134 | 0.228 | 0.150 | 0.251 | 0.310 | 0.356 | 0.223 | 0.280 | 0.188 | 0.318 | 0.210 | 0.283 | 0.211 | 0.278 | 0.215 | 0.280 |
| | 720 | 0.152 | 0.232 | 0.156 | 0.250 | **0.138** | **0.220** | 0.169 | 0.271 | 0.253 | 0.318 | 0.213 | 0.272 | 0.201 | 0.271 | 0.231 | 0.310 | 0.210 | 0.273 | 0.229 | 0.286 |
| | Avg | **0.129** | **0.203** | 0.142 | 0.238 | 0.132 | 0.220 | 0.154 | 0.255 | 0.263 | 0.322 | 0.232 | 0.291 | 0.194 | 0.276 | 0.211 | 0.285 | 0.221 | 0.287 | 0.233 | 0.295 |
| Exchange | 96 | **0.087** | **0.229** | 0.109 | 0.245 | 0.105 | 0.258 | 0.096 | 0.230 | 0.291 | 0.484 | 0.234 | 0.425 | 0.136 | 0.316 | 0.297 | 0.487 | 0.220 | 0.397 | 0.370 | 0.554 |
| | 192 | **0.147** | **0.298** | 0.197 | 0.350 | 0.258 | 0.393 | 0.205 | 0.340 | 0.236 | 0.421 | 0.231 | 0.430 | 0.186 | 0.360 | 0.300 | 0.461 | 0.153 | 0.334 | 0.427 | 0.590 |
| | 336 | **0.274** | **0.413** | 0.331 | 0.444 | 0.422 | 0.524 | 0.453 | 0.519 | 0.309 | 0.496 | 0.394 | 0.563 | 0.412 | 0.562 | 0.716 | 0.752 | 0.484 | 0.610 | 0.558 | 0.687 |
| | 720 | 0.338 | 0.470 | 0.362 | 0.482 | 0.552 | 0.577 | 0.419 | 0.502 | **0.244** | **0.419** | 0.390 | 0.558 | 0.459 | 0.614 | 0.387 | 0.559 | 0.336 | 0.511 | 0.503 | 0.640 |
| | Avg | **0.212** | **0.353** | 0.250 | 0.380 | 0.334 | 0.438 | 0.293 | 0.398 | 0.270 | 0.455 | 0.312 | 0.494 | 0.298 | 0.463 | 0.425 | 0.565 | 0.298 | 0.463 | 0.464 | 0.618 |
| 1st Count | | **28** | **28** | 2 | 3 | 2 | 1 | 1 | 2 | 3 | 4 | 0 | 0 | 1 | 0 | 0 | 0 | 2 | 1 | 1 | 1 |

