# OpenReview forum: "DAG: A Dual Correlation Network for Time Series Forecasting with Exogenous Variables"
_ICML.cc/2026/Conference — ICML 2026 regular_

### Official Review · Reviewer_eHrc · 2026-03-03

**Soundness:** 4
**Presentation:** 4
**Significance:** 4
**Originality:** 4
**Overall Recommendation:** 5
**Confidence:** 5

**Summary:**

This study proposes a general framework DAG, which utilizes dual correlation network along both the temporal and channel dimensions for time series forecasting with exogenous variables, especially leveraging future exogenous variable information. Extensive experiments on multiple datasets demonstrate the state-of-the-art performance of DAG.

**Compliance With Llm Reviewing Policy:**

Affirmed.

**Key Questions For Authors:**

Please refer to the Weaknesses.

**Limitations:**

Regarding future directions, the gating mechanism adopted in this work is relatively simple. Designing a more effective or expressive mechanism could potentially further enhance the overall performance of the model.

**Strengths And Weaknesses:**

Strengths:

S1. Multivariate time series forecasting with exogenous variables is important to various domains. S2. There are quite a few nice illustrations. S3. There is a detailed experimental comparison with many existing techniques.

S4. The core motivation is clear and well-defined, addressing a significant limitation of existing methods. The explicit modeling of how future covariates affect the target variable is of great practical significance.

S5. The model remains effective even when future exogenous variables are not available by using predicted values from its own discovery module.

Weaknesses:

W1. The author evaluated TSF-X on 12 new datasets but omitted standard multivariate forecasting benchmarks such as ETT. Testing on these widely recognized datasets would provide much stronger evidence of the algorithm's generalizability and effectiveness.

W2. Why was GCGNet, which is mentioned in the Related Work, excluded from the performance comparison? A direct comparison with such closely related methods is necessary to demonstrate the actual superiority of the proposed model.

W3. Why use a dot product of MLP-encoded series for the gating mechanism? Did you try other fusions? This is intended as a discussion point rather than a criticism.

---

> ### Author Rebuttal · Authors · 2026-03-30
>
> Dear Reviewer eHrc, thank you very much for your valuable suggestions and constructive attitude. We hope our response can address your concerns.
>
> **Reply to W1**
>
> We appreciate your valuable feedback. Our original focus was on forecasting tasks where future covariates are known, and the ETT datasets do not meet this requirement, which is why we initially did not evaluate on them. To more comprehensively assess the performance of DAG, we have now conducted experiments on standard benchmark datasets, including ETT, using only historical covariates for prediction. The experimental setup follows the protocol of Timexer. The results indicate that DAG maintains consistently strong performance across these benchmarks. Due to space limitations, please refer to the results at the following link: https://anonymous.4open.science/r/ICML-rebuttel-523D/ETT.md
>
> **Reply to W2**
>
> We thank the reviewer for pointing this out. To address this concern, we have now supplemented experiments including GCGNet in the performance comparison. The results confirm that DAG consistently outperforms GCGNet across the evaluated datasets, demonstrating the effectiveness and superiority of our proposed model.  Due to space limitations, please refer to the results at the following link: https://anonymous.4open.science/r/ICML-rebuttel-523D/GCGNet.md
>
> **Reply to W3 & Limitations**
>
> We appreciate the reviewer’s interest in the gating mechanism. We chose the dot product of MLP-encoded series for its simplicity and efficiency in capturing interactions between series. We also conducted preliminary experiments with alternative fusion methods (e.g., concatenation followed by MLP, element-wise addition). The dot product, however, provided a slightly better trade-off between accuracy and computational efficiency. In future work, we plan to explore more effective and fine-grained fusion mechanisms.

---

> > ### Author Rebuttal · Reviewer_eHrc · 2026-04-01
> >
> > Thanks to the author for answering my doubts. My concerns have been adequately addressed.

---

> > > ### Author Response · Authors · 2026-04-04
> > >
> > > Dear Reviewer, we are thrilled that our responses have effectively addressed your questions and comments. We would like to express our sincerest gratitude for taking the time to review our paper and provide us with such detailed feedback!

---

### Official Review · Reviewer_t1i8 · 2026-03-03

**Soundness:** 4
**Presentation:** 4
**Significance:** 4
**Originality:** 3
**Overall Recommendation:** 5
**Confidence:** 5

**Summary:**

This paper introduces a dual correlation network DAG for time series forecasting with exogenous variables. The authors argue that current methods don’t make full use of future exogenous variables and mostly ignore dependencies between endogenous and exogenous ones. The model is evaluated on standard benchmarks as well as several proprietary TSF-X datasets, and the authors report state-of-the-art results.

**Compliance With Llm Reviewing Policy:**

Affirmed.

**Final Justification:**

The comprehensive rebuttal and additional experiments effectively addressed my prior concerns regarding the method's soundness.

**Key Questions For Authors:**

Please refer to Paper Weaknesses. I will improve my score if my problem is solved.

**Limitations:**

I believe this work does not pose any apparent potential negative societal impacts. For areas that could be further improved, please refer to the Weaknesses.

**Strengths And Weaknesses:**

**Strengths**:

- Comprehensive experiments using multiple public datasets.

- This work focuses on an important problem that could have real-world applications.

- The paper is well-organized and clearly written. The motivation is clear. Figures and tables adequately support the text, making technical details accessible.



**Weaknesses**:

- Adapting non-exogenous methods via MLP-based fusion is beneficial; however, this strategy may not always be competitive, nor does it necessarily provide a faithful reflection of each method’s full potential. The discussion should therefore acknowledge the possible limitations or side effects introduced by this adaptation, in order to avoid over-attributing performance gains to the superiority of the DAG structure itself.

- The experimental results lack standard deviations or confidence intervals. Without these results, it is difficult to determine whether the reported performance gains are statistically significant.

- In section 5.3.1, replace future exogenous variables with predictor variables. Why choose this option instead of deleting the channel causal injection module?

- The methodology for determining the fusion and causality weights ($\lambda_1, \lambda_2$) requires further clarification. Specifically, were these hyperparameters fine-tuned individually for each dataset, or was a universal setting applied across all experiments?

---

> ### Author Rebuttal · Authors · 2026-03-30
>
> Dear Reviewer t1i8, thank you very much for your valuable feedback and for recognizing our work. We have carefully prepared the following responses and hope they address your concerns.
>
> **Reply to W1**
>
> Thank you for your valuable feedback! The original paper did not include a comparison between the original models and their versions augmented with the MLP fusion module in the forecasting scenario with exogenous variables. To demonstrate that our MLP fusion module effectively leverages exogenous variables and improves forecasting performance, we conducted additional experiments. We compared the original models, including DUET, Amplifier, TimeKAN, xPatch, and PatchTST, with their MLP-augmented versions. In addition, we evaluated two other fusion modules designed to enable traditional forecasting models to incorporate future exogenous variables: a convolution-based module and a cross-attention-based module. Specifically, given historical endogenous variables $X^{\text{endo}} \in \mathbb{R}^{N \times T}$ and historical exogenous variables $X^{\text{exo}} \in \mathbb{R}^{D \times T}$, the backbone model first generates a latent representation:
>
> $$z = \theta_{\text{Model}}(X^{\text{endo}}, X^{\text{exo}}),$$
>
> where $z \in \mathbb{R}^{N \times F}$ and $\theta_{\text{Model}}$ denotes the parameters of the forecasting model.
>
> The latent representation $z$ is then combined with the future exogenous variables $Y^{\text{exo}} \in \mathbb{R}^{N \times F}$ and processed by either:
>
> 1. **Conv Fusion Module**: a 1D convolution layer captures local correlations across channels, and the resulting feature is fused with the future exogenous variables to produce the final prediction.
>
> $$\hat{Y}^{\text{endo}} = \theta_{\text{Conv}}\big(\text{Concat}(z, Y^{\text{exo}})\big),$$
>
> where $\theta_{\text{Conv}}$ denotes the parameters of the convolution-based fusion module.
>
> 2. **Cross-Attention Fusion Module**: a cross-attention layer captures global dependencies between the future exogenous variables and the latent representation to produce the final prediction
>
> $$\hat{Y}^{\text{endo}} = \theta_{\text{CA}}\big(z, Y^{\text{exo}}\big),$$
>
> where $\theta_{\text{CA}}$ denotes the parameters of the cross-attention fusion module.
>
> The results show that our MLP fusion module consistently improves performance and is an effective method for using exogenous variables. The other two fusion modules were also tested in detail, and the MLP module achieved the best results, so we selected it as the fusion method. Due to space limitations, please refer to the results at the **following link**: https://anonymous.4open.science/r/ICML-rebuttel-523D/baseline.md
>
> **Reply to W2**
>
> We thank the reviewer for this important suggestion. To further substantiate the robustness of our results, we have supplemented additional experiments across multiple random seeds, and report the mean and standard deviation of the performance metrics. The results show consistently low variance, confirming the stability and reliability of DAG. We will include these statistics in the revised manuscript. Due to space limitations, please refer to the results at the **following link**: https://anonymous.4open.science/r/ICML-rebuttel-523D/stability_analysis.md
>
> **Reply to W3**
>
> We thank the reviewer for this valuable question. To better address this concern, we conducted additional experiments where future covariate information is not used.
>
> Specifically, we compared two strategies: (1) replacing future exogenous variables with predictor variables, and (2) directly removing the channel causal injection module. The experimental results show that replacing future exogenous variables with predictor variables consistently achieves better performance than deleting the channel causal injection module.
>
> This suggests that preserving the module structure while providing weaker or indirect auxiliary signals is more beneficial than completely removing the causal modeling mechanism. Due to space limitations, please refer to the results at the **following link**: https://anonymous.4open.science/r/ICML-rebuttel-523D/ablation_wo_future.md
>
> **Reply to W4**
>
> We appreciate the reviewer’s question regarding the fusion and correlation weights. In our experiments, the fusion weight (λ₁) and correlation weight (λ₂) were set individually for each dataset. This is because the degree of association between covariates and endogenous variables varies across datasets, and using a unified global setting could lead to suboptimal performance.

---

> > ### Author Rebuttal · Reviewer_t1i8 · 2026-04-02
> >
> > I thank the authors for the thorough rebuttal and the comprehensive additional experiments, which have effectively addressed all my previous concerns. Accordingly, I am raising my score and recommend incorporating these new results into the camera-ready version.

---

> > > ### Author Response · Authors · 2026-04-04
> > >
> > > Dear Reviewer, we are thrilled that our responses have effectively addressed your questions and comments. We would like to express our sincerest gratitude for taking the time to review our paper and provide us with such detailed feedback!

---

### Official Review · Reviewer_ANB8 · 2026-03-05

**Soundness:** 4
**Presentation:** 4
**Significance:** 4
**Originality:** 4
**Overall Recommendation:** 6
**Confidence:** 5

**Summary:**

This paper discusses the time series forecasting problem with exogenous variables (TSF-X), emphasizing the importance of effectively utilizing historical and future exogenous variables. They propose DAG, a dual correlation network with dedicated modules for modeling and injecting time and channel correlations between endogenous and exogenous variables. This architecture explicitly separates correlation discovery and injection, and utilizes attention based mechanisms to handle time and channel dependencies, combined with gating for adaptive fusion. Extensive experiments on 12 real-world datasets have shown that DAG outperforms TSF-X's recent strong baseline, and ablation confirms the importance of time and channel related modules.

**Compliance With Llm Reviewing Policy:**

Affirmed.

**Final Justification:**

Thank you for the clear and detailed rebuttal. My main concerns have been addressed. I am more confident in the work and am happy to increase my score.

**Key Questions For Authors:**

1. The authors conduct both short-term and long-term forecasting experiments. However, it remains unclear whether the model parameters are kept consistent across these two settings. This point should be clarified to ensure fairness and reproducibility of the experimental comparisons.

2. Have you evaluated the baseline under identical settings in artificially segmented endogenous/exogenous datasets?

3.  How the DAG implements the prediction of endogenous variables using only historical covariates is currently insufficient？

If the authors adequately address and resolve the concerns I have raised, I would be willing to reconsider and increase my rating accordingly.

**Limitations:**

Please see the weaknesses and questions.

**Strengths And Weaknesses:**

**Strengths:**

1. The DAG framework introduces a dual correlation modeling approach, handling both temporal and channel dependencies between exogenous and endogenous variables in TSF-X, which is an underexplored and impactful direction.

2. Empirical results (notably Table 1 and the extensive experiments in Appendix B.1) exhibit consistent improvements over a wide suite of baselines, including both models with and without native support for exogenous variables.

3. The released datasets (especially if truly future-known) could benefit the community.

4. The figures and tables used in this work are clear and easy to read.

**Weaknesses:**

1. The description of how the DAG implements the prediction of endogenous variables using only historical covariates is currently insufficient. Providing more technical details or a schematic would significantly aid reader understanding.

2. The current baseline includes several models that do not inherently support exogenous variables. Given that this paper focuses on leveraging exogenous information, the fairness and relevance of comparing against these "exogenous-blind" models should be addressed.

3. Have you evaluated the baseline under identical settings in artificially segmented endogenous/exogenous datasets? Please clarify.

---

> ### Author Rebuttal · Authors · 2026-03-30
>
> Dear Reviewer ANB8, we sincerely thank you for the valuable feedback and positive attitude. We hope our responses can address your other concerns and expect further discussions.
>
> **Reply to W1 & Q3**
>
> We thank the reviewer for the question. As described in Section 5.3.1 of the paper, we provide a more detailed explanation here. During training and validation, the model architecture remains unchanged. However, during inference, we use the predicted $\hat{Y}^{\text{exo}}$ from $\mathcal{F}{\theta_1}$ to replace the actual $Y^{\text{exo}}$ in $\mathcal{G}{\theta_4}$ when predicting the endogenous variables $Y^{\text{endo}}$. This approach effectively addresses the issue of unavailable future exogenous variables while allowing the model to predict endogenous variables based solely on historical covariates.
>
> **Reply to W2**
>
> Thank you for your valuable feedback! Since there are relatively few algorithms that natively support exogenous variables, we have evaluated a range of models designed for this setting, including TimeXer, TFT, TiDE, and CrossLinear. To provide a broader comparison, we also tested models that do not natively support exogenous variables. For these models, instead of applying them directly, we employed an MLP fusion module to enable them to effectively utilize exogenous information, which is described in detail of the original paper (Appendix A.3)
>
> **Reply to W3 & Q2**
>
> We thank the reviewer for the question. To rigorously validate our method, we evaluated all baseline models under identical settings on the same datasets. In Table 1, the inputs for all models include historical endogenous variables, historical exogenous variables, and future exogenous variables; for models that do not natively support exogenous variables, we reasonably extended them to accommodate such inputs. In Table 2, the inputs for all models are consistently set as historical endogenous and exogenous variables. The results demonstrate that our method consistently maintains significant performance advantages under this strict comparison.
>
> **Reply to Q1**
>
> We thank the reviewer for the question. We conduct both short-term and long-term forecasting experiments. For the Colbun and Raperl datasets, short-term forecasting uses a lookback window of 60 and a prediction horizon of 10, while long-term forecasting uses a lookback window of 180 and a prediction horizon of 30. For the other datasets, short-term forecasting uses a lookback window of 168 and a prediction horizon of 24, whereas long-term forecasting uses a lookback window of 720 and a prediction horizon of 360. Except for the lookback window and prediction horizon, all other model parameters are kept identical across short-term and long-term settings, and the DAG as well as all baseline models are configured consistently, ensuring fairness in the experimental comparisons.

---

> > ### Author Rebuttal · Reviewer_ANB8 · 2026-04-03
> >
> > Thank you for the clear and detailed rebuttal. My main concerns have been addressed. I am more confident in the work and am happy to increase my score.

---

> > > ### Author Response · Authors · 2026-04-04
> > >
> > > Dear Reviewer, we are thrilled that our responses have effectively addressed your questions and comments. We would like to express our sincerest gratitude for taking the time to review our paper and provide us with such detailed feedback!

---

### Official Review · Reviewer_6J8b · 2026-03-11

**Soundness:** 4
**Presentation:** 3
**Significance:** 4
**Originality:** 4
**Overall Recommendation:** 5
**Confidence:** 5

**Summary:**

The paper proposes DAG, a dual correlation network for time series forecasting with exogenous variables (TSF-X) that aims to explicitly leverage both historical and future covariates.  In 12 TSF-X datasets (including 5 EPF datasets and 7 newly collected datasets), the model reports strong performance compared to baseline, with ablation indicating benefits for both time and channel related pathways.

**Compliance With Llm Reviewing Policy:**

Affirmed.

**Final Justification:**

Rebuttal addressed my concerns, and I would remain my suggestion to accept the paper.

**Key Questions For Authors:**

Kindly help to address the weakness points mentioned above.

Q1 Could the authors include additional experiments on the ETT datasets? As these are standard benchmarks in the field of long-term time series forecasting, their inclusion would significantly strengthen the validation of the proposed method.

Q2 Can you provide standard deviations (across multiple seeds/runs) for your results to better evaluate statistical significance?

Q3 Can you analyze the interaction between these two modules in more detail? For example, is there a specific dataset where one module contributes more than the other? If so, why?

**Limitations:**

This paper does not include the limitations sections. My suggestions are as follow:

For future work, it would be promising to transition from correlation-based modeling to causal inference. Explicitly modeling the causal relationships between covariates and endogenous variables could further enhance both the interpretability and the reliability of the model. Additionally, I do not foresee any negative social impacts arising from this work.

**Strengths And Weaknesses:**

Strengths:

S1. TSF-X with known future covariates is practically important (e.g., energy, retail, traffic). A principled mechanism to exploit future exogenous inputs beyond naive concatenation is valuable. Unlike previous methods that underutilize exogenous data or rely on simple combinations of inputs, DAG explicitly models relationships along two critical axes.
S2. The overall structure of the paper is logical and easy to follow, clearly presenting the motivation and methodology.
S3. The experimental validation is solid, including 12 datasets. The establishment of new TSF-X benchmarks effectively demonstrates the method's effectiveness.

Weaknesses:

W1. Some references in the manuscript remain in arXiv form and need to be updated to their final published versions.

W2. The experimental evaluation lacks results on the ETT datasets, which are widely considered the standard benchmarks in the field of long-term time series forecasting.

W3. Can you provide standard deviations (across multiple seeds/runs) for your results to better evaluate statistical significance?

W4.  Can you analyze the interaction between these two modules in more detail? For example, is there a specific dataset where one module contributes more than the other? If so, why?

---

> ### Author Rebuttal · Authors · 2026-03-30
>
> Dear Reviewer 6J8b, thank you very much for your valuable feedback and for recognizing our work. We have carefully prepared the following responses and hope they address your concerns.
>
> **Reply to W1**
>
> We will update all outdated references by replacing ArXiv versions with their corresponding official conference or journal publications in the revised manuscript.
>
> **Reply to W2 & Q1**
>
> We appreciate your valuable feedback. Our original focus was on forecasting tasks where future covariates are known, and the ETT datasets do not meet this requirement, which is why we initially did not evaluate on them. To more comprehensively assess the performance of DAG, we have now conducted experiments on standard benchmark datasets, including ETT, using only historical covariates for prediction. The experimental setup follows the protocol of Timexer. The results indicate that DAG maintains consistently strong performance across these benchmarks. Due to space limitations, please refer to the results at the **following link**:
> https://anonymous.4open.science/r/ICML-rebuttel-523D/ETT.md
>
> **Reply to W3 & Q2**
>
> We thank the reviewer for this important suggestion. To further substantiate the robustness of our results, we have supplemented additional experiments across multiple random seeds, and report the mean and standard deviation of the performance metrics. The results show consistently low variance, confirming the stability and reliability of DAG. We will include these statistics in the revised manuscript. Due to space limitations, please refer to the results at the **following link**:
> https://anonymous.4open.science/r/ICML-rebuttel-523D/stability_analysis.md
>
> **Reply to W4 & Q3**
>
> We thank the reviewer for the insightful question. The two modules in DAG are complementary. Specifically, the Temporal Correlation Module (TCM) captures how historical exogenous variables affect future exogenous variables and incorporates these correlations into forecasting future endogenous variables based on historical endogenous variables. The Channel Correlation Module (CCM), on the other hand, models how historical exogenous variables influence historical endogenous variables, enhancing the prediction of future endogenous variables based on future exogenous variables. From the ablation study (Table 3), where “d” and “e” correspond to these two modules, we observe that removing either module leads to performance degradation. Notably, the relative contribution of each module varies across datasets: removing one module does not consistently result in worse performance than removing the other. This behavior mainly depends on the degree to which exogenous variables influence endogenous variables in a given dataset. Furthermore, in our parameter sensitivity analysis, we examine the fusion weights for combining the predictions of these two modules (Figure 5a), which provides additional insight into their complementary roles and relative importance across datasets.
>
> **Reply to Limitations**
>
> We thank the reviewer for the valuable suggestion. We agree that explicitly modeling causal relationships between covariates and endogenous variables is a promising direction for future work. While the current DAG framework focuses on capturing correlations, extending it toward causal inference could further improve both interpretability and reliability, which we plan to explore in our future research.

---

> > ### Author Rebuttal · Reviewer_6J8b · 2026-04-01
> >
> > Thanks to the author for addressing my questions. Since my original rating is accept, I will not further change the score.

---

> > > ### Author Response · Authors · 2026-04-04
> > >
> > > Dear Reviewer, we are thrilled that our responses have effectively addressed your questions and comments. We would like to express our sincerest gratitude for taking the time to review our paper and provide us with such detailed feedback!

---

### Decision · Program_Chairs · 2026-04-30

**Decision:**

Accept (regular)

**Comment:**

This paper proposes a new neural network architecture for time-series forecasting with exogenous variables. The key observation is that existing methods neither (i) exploit the forecasted exogenous variables nor (ii) explicitly account for the correlation between endogenous and exogenous variables. By developing specialized modules to address these two limitations, the proposed approach can effectively improve forecasting performance.
All reviewers acknowledge the authors’ rebuttal and note that the authors effectively addressed several concerns, including providing additional experiments on the ETT datasets, reporting standard deviations, and adding more baselines. Given the reviewers’ positive consensus, I am pleased to recommend acceptance of this manuscript.